# Understanding Generalization in Physics Informed Models through Affine Variety Dimensions

Takeshi Koshizuka[1] and Issei Sato[1]

[1]Department of Computer Science, The University of Tokyo
[1]{koshizuka-takeshi938444, sato}@g.ecc.u-tokyo.ac.jp

## Abstract

Physics-informed machine learning is gaining significant traction for enhancing statistical performance and sample efficiency through the integration of physical knowledge. However, current theoretical analyses often presume complete prior knowledge in non-hybrid settings, overlooking the crucial integration of observational data, and are frequently limited to linear systems, unlike the prevalent nonlinear nature of many real-world applications. To address these limitations, we introduce a unified residual form that unifies collocation and variational methods, enabling the incorporation of incomplete and complex physical constraints in hybrid learning settings. Within this formulation, we establish that the generalization performance of physics-informed regression in such hybrid settings is governed by the dimension of the affine variety associated with the physical constraint, rather than by the number of parameters. This enables a unified analysis that is applicable to both linear and nonlinear equations. We also present a method to approximate this dimension and provide experimental validation of our theoretical findings.

## 1 Introduction

In recent years, physics-informed machine learning (PIML) has garnered significant attention [35, 25, 12, 19]. PIML represents a hybrid approach that integrates physical knowledge into machine learning models for tasks involving physical phenomena. These hybrid models can leverage inherent physical structures, such as differential equations [36], conservation laws [23], and symmetries [3], as inductive biases. This integration has the potential to enhance both sample efficiency and generalization capabilities. These models have been empirically applied to a wide range of phenomena, with successful applications observed in areas including thrombus material properties [41], fluid dynamics [8, 24], turbulence [40], and heat transfer problems [9].

Despite these empirical successes, the theoretical analysis of PIML remains underdeveloped, which potentially undermines the reliability of these methods. Notably, in practical scenarios, prior knowledge of the governing differential equations, particularly their source terms or boundary conditions, is often incomplete. Consequently, learning frequently involves a hybrid approach of fitting to actual observational data alongside incorporating physical constraints. However, much of the existing theoretical research focuses on settings where complete prior knowledge of the differential equations is assumed. Furthermore, the scope of existing analyses is often limited to linear differential equations or systems exhibiting strong regularity, creating a gap between theory and application.

To bridge this gap, we propose a versatile analytical framework for physics-informed regression using linear hypothesis classes over nonlinear features in hybrid settings. Our key idea is to formulate the differential equation constraints by introducing a Unified Residual Form. This form, defined on a finite set of trial functions and a measure, provides a practical approximation of physical constraints. This formulation unifies the collocation-based constraints, used in Physics-Informed Neural Networks

(PINNs) [36], and the standard unified residual form constraints, used in the variational and finite element methods. Within this framework, the learning weights of a physics-informed linear regressor are shown to be defined on an affine variety associated with the differential equation. Crucially, our analysis aims to elucidate the impact of incorporating physical prior knowledge on the generalization capacity of these models. We then establish that the generalization capacity of these models is determined by the dimension of this affine variety, rather than by the number of parameters. This novel perspective enables a unified analysis applicable to various equations, including nonlinear ones. To support our theoretical findings, we introduce a method for approximately calculating the dimension of the affine variety and provide extensive experimental validation. Our results illustrate that even in scenarios with a large number of parameters relative to the amount of data, the incorporation of physical structure reduces the intrinsic dimension of the hypothesis space, thereby mitigating overfitting and corroborating our theoretical claims.

## 2 Related Work

Since the seminal work by Raissi et al. [36] on PINNs, PIML has rapidly emerged as a significant field of study. This area has been comprehensively surveyed in the literature by [35, 25, 12, 19]. Leveraging the high function approximation capabilities of neural networks [20, 26, 14], these models have been employed as versatile surrogates for solving various equations. In contrast, linear models are also used because of their interpretability, consistency with classical numerical solvers [5, 18], and the close relationship between Partial Differential Equations (PDEs) and kernel methods [37, 11, 27, 13, 16]. Recently, methods that exploit underlying conservation laws [23, 21] and symmetries [3, 13], in addition to the equations themselves, have also been developed.

Recent studies have made advances in the theoretical understanding of PINNs. Shin [38] rigorously showed that the minimizer of the PINN loss converges to the strong solution as the data size approaches infinity for linear elliptic and parabolic PDEs under certain conditions. These findings were extended by Shin et al. [39] into a general framework applicable to broader linear problems, with the loss function formulated in both strong and variational forms. Mishra and Molinaro [30, 31] use the stability properties of the underlying PDEs to derive upper bounds on the generalization error of PINNs. Subsequent research has applied this analytical framework to various specific equations [6, 29]. However, studies explicitly addressing the impact of physical structure on generalization capabilities are still limited. Arnone et al. [5] proved that for second-order elliptic PDEs, the physics-informed linear estimator using a finite element basis converges at a rate surpassing the Sobolev minimax rate. Doumèche et al. [15] quantified the generalization capacity of the physics-informed estimator for general linear PDEs using the concept of effective dimension [10], a well-known metric in the analysis of the kernel method. The effects of incorporating the structures of nonlinear complex equations, as well as conservation laws and symmetries, into models on generalization, have yet to be thoroughly analyzed.

## 3 Minimax risk Analysis

In this section, we explain how introducing physical structures can improve the generalization capacity of linear models. In Section 3.1, we outline the problem setup. In Section 3.2, we perform a minimax risk analysis, showing that the generalization capacity is mainly determined by the dimension of the affine variety. In Section 3.3, we show that our theory aligns with existing theories on linear operators. Notations are summarized in Appendix A.

### 3.1 Problem Setup

We consider the regression problem, which aims to learn the unknown function $f^*\colon \mathbb{R}^m \to \mathbb{R}$ that satisfies the differential equation. We have a dataset consisting of $n$ observations, denoted as $\{(x_i, y_i)\}_{i=1}^n$, where $x_i \in \Omega \cup \partial\Omega$ represents the input within the domain $\Omega \subseteq \mathbb{R}^m$ or the boundary $\partial\Omega$ and $y_i \in \mathbb{R}$ represents the corresponding output. Observations are sampled independently from a probability distribution $\mathcal{P}$ on the domain $\Omega \cup \partial\Omega \times \mathbb{R}$. The relationship between the observations and the true function can be expressed as:

$$y_i = f^*(x_i) + \varepsilon_i, \ \varepsilon_i \sim \mathcal{N}(0, \sigma^2),$$

where $\varepsilon_i$ represents normally distributed noise with mean zero and variance $\sigma^2$. The target function $f^*$ is the solution of the differential equation, *i.e.*, $\mathscr{D}[f^*] = 0$ for a given operator $\mathscr{D}\colon L^2(\Omega) \to L^2(\Omega)$, where $L^2(\Omega)$ denotes the space of square-integrable functions on a domain $\Omega \subseteq \mathbb{R}^m$. For a more detailed background on the problem setting, please refer to Appendix C.

**Unified Residual Form:** To formulate prior knowledge of the governing differential equations, we first introduce a unified residual form, which captures physical constraints in an integrated or averaged sense. Such formulations naturally arise in variational and finite element methods, and are particularly well-suited to hybrid settings where only partial physical supervision is available. Formally, let $\mathcal{T} := \{(\psi_k, \mu_k)\}_{k=1}^K$ be a finite collection of trial functions and measure pairs, where each $\psi_k : \mathbb{R}^m \to \mathbb{R}$ is a smooth trial function and $\mu_k : \Sigma \to \mathbb{R}$ is a measure on the $\sigma$-algebra $\Sigma$ over the domain $\Omega$. Then, the unified residual form of the knwon differential equation $\mathscr{D}$ is defined by

$$\langle \mathscr{D}[f], \psi_k \rangle_{\mu_k} := \int_\Omega \mathscr{D}[f](x)\,\psi_k(x)\,\mathrm{d}\mu_k(x) = 0, \quad k = 1, \ldots, K.$$

We then impose the differential equation constraint through the unified residual form defined above. The resulting physics-informed regression problem reads:

$$\hat{f}_n = \underset{f \in \mathcal{F}(\mathscr{D}, \mathcal{T})}{\arg\min} \frac{1}{n} \sum_{i=1}^n |y_i - f(x_i)|^2 + \lambda_n \|f\|^2,$$

$$\mathcal{F}(\mathscr{D}, \mathcal{T}) := \{ f : \langle \mathscr{D}[f], \psi_k \rangle_{\mu_k} = 0, \ \forall (\psi_k, \mu_k) \in \mathcal{T} \},$$
(1)

where $\lambda_n$ is a regularization parameter and $\|\cdot\|$ denotes the standard $L^2$ norm. This formulation relaxes the classical smoothness requirements while still leveraging physics-informed constraints via a unified measure-based approach: choosing Borel measures leads to an approximation of standard weak solutions, whereas choosing Dirac measures leads to an approximation of the strong-form residuals used in the PINN framework.

**Physics-Informed Linear Regression (PILR) Setup:** Let $\mathcal{B} = \{\phi_j : \mathbb{R}^m \to \mathbb{R}\}_{j=1}^d$ be a fixed basis. Define the basis vector $\boldsymbol{\phi}(x) = [\phi_1(x), \phi_2(x), \ldots, \phi_d(x)]^\top \in \mathbb{R}^d$, the design matrix $\boldsymbol{\Phi} = [\boldsymbol{\phi}(x_1), \boldsymbol{\phi}(x_2), \ldots, \boldsymbol{\phi}(x_n)]^\top \in \mathbb{R}^{n \times d}$, and the target vector $\boldsymbol{y} = [y_1, \ldots, y_n]^\top \in \mathbb{R}^n$.

The physics-informed feasible set is

$$\mathcal{V}(\mathscr{D}, \mathcal{B}, \mathcal{T}) := \left\{ \boldsymbol{w} \in \mathbb{R}^d : \langle \mathscr{D}\left[\boldsymbol{w}^\top \boldsymbol{\phi}\right], \psi_k \rangle_{\mu_k} = 0, \ \forall (\psi_k, \mu_k) \in \mathcal{T}, \phi_j \in \mathcal{B} \right\}.$$
(2)

The problem Eq. (1) reduces to the physics-informed linear regression given by

$$\widehat{\boldsymbol{w}} = \underset{\boldsymbol{w} \in \mathcal{V}_R}{\arg\min} \frac{1}{n} \|\boldsymbol{y} - \boldsymbol{\Phi}\boldsymbol{w}\|_2^2,$$
(3)

where $\mathcal{V}_R = \mathcal{V}(\mathscr{D}, \mathcal{B}, \mathcal{T}) \cap \mathbb{B}_2(R)$ is the affine variety constrained by the $\ell_2$-ball $\mathbb{B}_2(R)$ with radius $R > 0$ and $\|\cdot\|_2$ is the $\ell_2$-norm.

The set of coefficients $\mathcal{V}$ constitutes *an affine variety* as it represents the set of solutions to the $K$ polynomial equations in the $d$ variables with real coefficients. For example, when $m = 1$ and $\mathscr{D}[f] = f \cdot \frac{\mathrm{d}}{\mathrm{d}x} f$, the affine variety $\mathcal{V}$ is defined by the solution set of the polynomial equations $p_k(\boldsymbol{w}) = \sum_{j,j'=1}^d \langle (\frac{\mathrm{d}}{\mathrm{d}x} \phi_j)\,\phi_{j'}, \psi_k \rangle_{\mu_k} w_j w_{j'} = 0$ for $k = 1, \ldots, K$. We perform minimax risk analysis based on the dimension $d_\mathcal{V}$ of this affine variety because the affine variety $\mathcal{V}$ is crucial to determine the size of the intrinsic hypothesis space.

**Minimax risk:** The goal of our analysis is to obtain the upper bound of the minimax risk for PILR in Eq. (3), which is defined by

$$\min_{\hat{\boldsymbol{w}}} \max_{\boldsymbol{w}^* \in \mathcal{V}_R} \|\hat{\boldsymbol{w}} - \boldsymbol{w}^*\|_2^2,$$
(4)

Here, $\boldsymbol{w}^* \in \mathcal{V}_R$ represents the optimal weight vector. The corresponding optimal hypothesis $f_{\boldsymbol{w}^*} = \boldsymbol{w}^{*\top}\boldsymbol{\phi}$ within our hypothesis space $\mathcal{H} = \{\boldsymbol{w}^\top \boldsymbol{\phi} : \boldsymbol{w} \in \mathcal{V}_R\}$ is defined as the best approximation of the true function $f^*$: $f_{\boldsymbol{w}^*} = \boldsymbol{w}^{*\top}\boldsymbol{\phi} = \arg\min_{f_{\boldsymbol{w}} \in \mathcal{H}} \|f_{\boldsymbol{w}} - f^*\|^2$.

**We strongly recommend referring to the example in Section 5.1 to intuitively understand our problem setting.**

## 3.2 Main Theorem

In this section, we present an upper bound on the minimax risk for PILR. The bound is interpretable and sufficiently sharp, revealing how physical constraints reduce hypothesis complexity and enhance generalization. We begin by stating the definition and assumptions underpinning our analysis.

**Definition 3.1** $((\beta, d_\mathcal{V})$-regular set). An affine variety $V \subseteq \mathbb{R}^d$ is called a $(\beta, d_\mathcal{V})$-regular set if the following conditions hold: (1) For almost all affine subspaces $L \subseteq \mathbb{R}^d$ of dimension $d_L$ satisfying $d - d_L \leq d_\mathcal{V}$, the intersection $V \cap L$ has at most $\beta$ path-connected components. (2) For almost all affine subspaces $L \subseteq \mathbb{R}^d$ of dimension $d_L$ with $d - d_L > d_\mathcal{V}$, the intersection $V \cap L$ is empty. See Appendix B.2 for illustrative explanations.

**Assumption 3.2** (Boundedness of basis functions). For the basis function $\phi = [\phi_1, \ldots, \phi_d]^\top$, where $\phi_j \in \mathcal{B}$, assume that there exists a positive constant $M$ such that $\|\phi(x)\|_2 \leq M$ for all $x \in \Omega$.

**Assumption 3.3.** Assume there exists a constant $\eta > 0$ such that $\frac{1}{\sqrt{n}}\|\Phi w\|_2 \geq \sqrt{\eta}\|w\|_2$ for all $w \in \mathbb{B}_2(2R)$.

**Assumption 3.4** (Stability of estimator). Assume there exists a constant $\Gamma > 1$ such that $\|\hat{w}_1 - \hat{w}_2\|_2 \leq (\Gamma - 1)\|w_1^* - w_2^*\|_2$, for the estimators $\hat{w}_1$ and $\hat{w}_2$ of the optimal weights $w_1^*$ and $w_2^*$, respectively.

Next, we present the upper bound on the minimax risk. The complete proof is provided in Appendix D.

**Theorem 3.5** (Minimax Risk Bound). *Let $\mathcal{V}(\mathcal{D}, \mathcal{B}, \mathcal{T})$ be the $(\beta, d_\mathcal{V})$-regular affine variety defined in Eq.* (2). *Suppose Assumptions 3.2-3.4 hold. Then, there exists a positive constant $C$, independent of $n$, $d_\mathcal{V}$, $d$, and $\beta$, such that for any $\delta \in (0, 1)$, with probability at least $1 - \delta$, the minimax risk for PILR defined by Eq.* (4) *is bounded by*

$$\min_{\hat{w}} \max_{w^* \in \mathcal{V}_R} \|\hat{w} - w^*\|_2^2 \leq C\eta^{-1}\sigma M \Gamma R \left( \sqrt{\frac{d_\mathcal{V} \log(d_\mathcal{V} d)}{n}} + \sqrt{\frac{\log 2\beta}{n}} + 2\sqrt{\frac{\log(2/\delta)}{n}} \right). \quad (5)$$

*Proof Sketch.* The proof proceeds in two steps. In the first step, we upper bound the minimax risk by the supremum of a sub-Gaussian random process defined over the metric space $(\mathcal{V}_R, \|\cdot\|_2)$. The second step utilizes Dudley's integral theorem, which bounds the supremum of the process by an integral involving its covering number, specifically: $\int_0^\infty \sqrt{\mathcal{N}(\mathcal{V}_R, \varepsilon, \|\cdot\|_2)}\,\mathrm{d}\varepsilon$. To apply Dudley's theorem effectively, we employ Lemma B.2 to obtain an explicit upper bound for the covering number. Substituting this bound into Dudley's integral and performing the integration yields the desired high-probability minimax risk bound. $\qquad \square$

Theorem 3.5 demonstrates that the minimax risk is primarily governed by the intrinsic dimension $d_\mathcal{V}$ of the affine variety $\mathcal{V}$, rather than the ambient input dimension $d$, particularly when the topological complexity parameter $\beta$ is small. For comparison, standard least-squares estimation over an $\ell_2$-ball $\mathbb{B}_2(R) \subset \mathbb{R}^d$ yields a minimax risk rate of order $\mathcal{O}(\sqrt{d/n})$, which is optimal for unconstrained linear regression in $d$-dimensional space. In contrast, our result shows that when $d_\mathcal{V} \ll d$, incorporating physical structure into the hypothesis space through differential constraints significantly sharpens the risk rate, yielding improved generalization.

**On the Role of $\beta$.** The parameter $\beta$ captures the topological complexity of the affine variety and appears as a regularity constant in the generalization bound. Its upper bound can be estimated via the Petrovskii–Oleinik–Milnor inequality [33, 32, 28], which provides a bound on the sum of Betti numbers of a semialgebraic set. Specifically, if the variety $\mathcal{V} \cap \mathbb{B}_2(R) \subset \mathbb{R}^d$ is defined by polynomial constraints $\{p_k(w)\}_{k=1}^K$ of maximal degree $\rho$, then it is $(\rho(2\rho - 1)^{d+1}, d_\mathcal{V})$-regular. This implies that as the degree $\rho$ of the defining polynomials increases, the variety can exhibit more intricate topological features, such as additional holes and disconnected components.

**How $d_\mathcal{V}$ and $\beta$ Arise from the Covering Argument.** The minimax risk is bounded via Dudley's entropy integral, which requires control over the covering number $\mathcal{N}(\mathcal{V}_R, \varepsilon, \|\cdot\|_2)$. Following the geometric approach of Zhang and Kileel [42], the affine variety $\mathcal{V} \subset \mathbb{R}^d$ is sliced using a family of

linear subspaces $\{L_s\}_{s\in\mathbb{N}}$, and each intersection $\mathcal{V} \cap L_s$ is covered by Euclidean balls of radius $\varepsilon$. The total covering is then given by

$$\mathcal{V} \subset \bigcup_s \bigcup_{v \in \mathcal{V} \cap L_s} \mathbb{B}_2(v; \varepsilon).$$

In this construction, the intrinsic dimension $d_{\mathcal{V}}$ controls the number of subspaces required to sufficiently cover $\mathcal{V}$, while the parameter $\beta$, corresponding to the sum of Betti numbers, governs the covering number of each individual section $\mathcal{V} \cap L_s$. Topologically, $\beta$ can be interpreted as quantifying the number of topological features (e.g., holes) in $\mathcal{V}$, and thus reflects the local geometric complexity encountered within each subspace. For reference, the standard covering number of the Euclidean ball satisfies $\mathcal{N}(\mathbb{B}_2(R), \varepsilon, \|\cdot\|_2) \leq (1 + 2R/\varepsilon)^d$, highlighting the advantage of replacing ambient-dimension dependence with complexity parameters intrinsic to the constraint set.

**Key Insights.** A central contribution of our analysis is its interpretability through the lens of intrinsic complexity measures. The dimension $d_{\mathcal{V}}$ plays a role analogous to the VC dimension in classification [1] or the pseudo-dimension in regression [34], serving as a proxy for the effective capacity of the hypothesis space. This dimensional viewpoint clarifies how the incorporation of physical constraints—via differential equation structure—can substantially reduce hypothesis complexity, even in high-dimensional ambient spaces. While this may come at the cost of slightly looser constants compared to minimax-optimal bounds, the resulting rate is still sharp enough to meaningfully capture the generalization benefit of physics-informed inductive bias. Empirical evidence supporting this theoretical advantage is presented in Section 5, and an alternative analysis via Rademacher complexity is provided in Appendix F.

**Effect of the Trial Function Set $\mathcal{T}$.** The set of trial functions $\mathcal{T}$ encodes the imposed physical constraints, typically derived from a governing differential operator $\mathscr{D}$. The cardinality $K = |\mathcal{T}|$ quantifies the amount of physical knowledge embedded in the learning problem. Increasing the number of trial functions leads to a more restrictive constraint set, which geometrically corresponds to a lower-dimensional affine variety. Specifically, if $\mathcal{T}_1 \subset \mathcal{T}_2$, then it follows that

$$\mathcal{V}(\mathscr{D}, \mathcal{B}, \mathcal{T}_2) \subset \mathcal{V}(\mathscr{D}, \mathcal{B}, \mathcal{T}_1) \quad \Rightarrow \quad d_{\mathcal{V}(\mathscr{D}, \mathcal{B}, \mathcal{T}_2)} \leq d_{\mathcal{V}(\mathscr{D}, \mathcal{B}, \mathcal{T}_1)},$$

which highlights how adding more physical constraints systematically reduces hypothesis complexity and improves generalization behavior.

### 3.3 Analysis on Linear Operator

We discuss the special case where $\mathscr{D}$ is a linear operator. The second term in Eq. (5) vanishes because the Petrovskii-Oleinik-Milnor inequality indicates $\beta = 1$. Thus, the minimax risk is $\mathcal{O}\left(\sqrt{d_{\mathcal{V}} \log(d_{\mathcal{V}} d)/n}\right)$. Furthermore, the affine variety $\mathcal{V}$ is the solution set of a homogeneous system of linear equations. That is, the affine variety can be written as $\mathcal{V}(\mathscr{D}, \mathcal{B}, \mathcal{T}) = \{\boldsymbol{w} : \boldsymbol{D}\boldsymbol{w} = \boldsymbol{0}\}$ using the matrix $\boldsymbol{D} \in \mathbb{R}^{K \times d}$ defined by $D_{k,j} := \langle \mathscr{D}[\phi_j], \psi_k \rangle_{\mu_k}$. The affine variety is a linear subspace of dimension $d_{\mathcal{V}} = \dim \ker \boldsymbol{D}$. From the rank–nullity theorem, $d_{\mathcal{V}} = d - \operatorname{rank} \boldsymbol{D}$, indicating that the higher the rank of the matrix $\boldsymbol{D}$, the better the minimax risk of regression.

We show that our theory is consistent with existing theories. The effect of incorporating physical structure, represented by linear differential equations, on generalization has been analyzed within the framework of kernel methods by Doumèche et al. [15, 16]. They argued that the physical structure smooths the kernel and reduces the effective dimension, leading to an improvement in the $\ell_2$ predictive error. We first present the definition of the physics-informed (PI) kernel.

**Definition 3.6** (PI kernel [15, 16]). Given a basis $\mathcal{B} = \{\phi_j\}_{j=1}^d$, trial functions (with a single measure) $\mathcal{T} = \{(\psi_k, \mu)\}_{k=1}^K$, and a linear operator $\mathscr{D}$, the *PI kernel* associated with the affine variety $\mathcal{V}(\mathscr{D}, \mathcal{B}, \mathcal{T}) = \{\boldsymbol{w} \in \mathbb{R}^d : \boldsymbol{D}\boldsymbol{w} = \boldsymbol{0}\}$ is defined as:

$$\kappa_{\boldsymbol{M}}(x, y) = \left\langle \boldsymbol{M}^{-1/2}\phi(x), \boldsymbol{M}^{-1/2}\phi(y) \right\rangle_2, \tag{6}$$

with

$$\boldsymbol{M}(\xi, \nu) := \xi\boldsymbol{I} + \nu\boldsymbol{D}^\top\boldsymbol{T}\boldsymbol{D}, \quad \boldsymbol{T}_{k,k'} = \langle \psi_k, \psi_{k'} \rangle_\mu, \quad \boldsymbol{D}_{k,j} = \langle \mathscr{D}[\phi_j], \psi_k \rangle_\mu.$$

Here, $\boldsymbol{I}$ is the identity matrix, and the matrix $\boldsymbol{T}$ is positive semi-definite. The parameters $\xi, \nu \geq 0$ control the balance between the $L^2$-regularization and the constraints derived from the operator $\mathscr{D}$.

Doumèche et al. [16] showed the effective dimension $d_{\text{eff}}(\xi, \nu)$ of the PI kernel is evaluated above by a computable quantity as follows:

$$d_{\text{eff}}(\xi, \nu) \lesssim \sum_{\alpha \in \sigma(\boldsymbol{B}\boldsymbol{M}^{-1}\boldsymbol{B})} \frac{1}{1 + \alpha^{-1}}, \tag{7}$$

where $\sigma(\cdot)$ denotes the spectrum (set of eigenvalues) of the given matrix, $\boldsymbol{B} \in \mathbb{R}^{d \times d}$ is the Gram matrix of the basis function, *i.e.*, $B_{j,j'} = \langle \phi_j, \phi_{j'} \rangle_\mu$. Next, we provide an explicit upper bound on the effective dimension of the PI kernel defined using the affine variety:

**Proposition 3.7.** *The effective dimension of the PI kernel associated with the affine variety* $\mathcal{V}(\mathscr{D}, \mathcal{B}, \mathcal{T}) = \{\boldsymbol{w} : \boldsymbol{D}\boldsymbol{w} = \boldsymbol{0}\}$ *with dimension* $d_\mathcal{V}$ *is upper bounded by*

$$d_{\text{eff}}(\xi, \nu) \lesssim \sum_{j=1}^{d_\mathcal{V}} \frac{1}{1 + \xi} + \sum_{j=d_\mathcal{V}}^{d} \frac{1}{1 + \xi + \nu \alpha_j} \leq \frac{d}{1 + \xi}.$$

*where* $\{\alpha_j\}_{j=d_\mathcal{V}}^{d}$ *denote the positive eigenvalues of the matrix* $\boldsymbol{D}^\top \boldsymbol{T} \boldsymbol{D}$.

Proposition 3.7 indicates that as the dimension of the affine variety $d_\mathcal{V} = d - \text{rank}\,\boldsymbol{D}$ decreases, the upper bound on the effective dimension of the PI kernel decreases accordingly. Since the matrix $\boldsymbol{D}^\top \boldsymbol{T} \boldsymbol{D}$ is positive semi-definite, all eigenvalues satisfy $\alpha_j \geq 0$. The intrinsic dimension $d_\mathcal{V}$ corresponds precisely to the number of zero eigenvalues ($\alpha_j = 0$). The terms in the second sum ($j > d_\mathcal{V}$) involve strictly positive eigenvalues $\alpha_j > 0$. Given that $\nu > 0$, we have $\frac{1}{1+\xi+\nu\alpha_j} < \frac{1}{1+\xi}$. Thus, when the intrinsic dimension $d_\mathcal{V}$ decreases, the number of terms in the first sum (with the larger value $1/(1 + \xi)$) decreases, while the number of terms in the second sum (with smaller values $1/(1 + \xi + \nu \alpha_j)$) increases. This shift towards smaller-valued terms leads to an overall reduction in the complexity bound.

Consequently, our theoretical results align with the existing PI kernel theory [15, 16]. The PI kernel framework from the previous literature quantifies the complexity of the hypothesis space through the entire spectrum of the matrix $\boldsymbol{D}$ combined with the base kernel $\langle \phi(x), \phi(y) \rangle_2$, restricting the analysis primarily to linear target operators $\mathscr{D}$. In contrast, our approach allows for analysis of linear and non-linear operators by focusing solely on the intrinsic dimension $d_\mathcal{V}$ (the count of zero eigenvalues), rather than analyzing the entire eigenvalue spectrum.

## 4 On the Dimension of an Affine Variety

In general, the dimension of the affine variety $V = \{\boldsymbol{w} \in \mathbb{R}^d : p_k(\boldsymbol{w}) = 0, \forall k = 1, \ldots, K\}$ defined by polynomials $\{p_k\}_{k=1}^{K}$ has many equivalent definitions. In particular, the following statements are all equivalent.

**Definition 4.1.** The maximal length of the chains $V_0 \subset V_1 \subset \ldots \subset V_{d_V}$ of non-empty subvarieties of $V$.

**Definition 4.2.** The degree of the denominator of the Hilbert series of the affine variety $V$.

**Definition 4.3.** The maximal dimension of the tangent vector spaces at the non-singular points $U \subseteq V \subset \mathbb{R}^d$ of the variety, *i.e.*, $d_V = \max_{\boldsymbol{w} \in U} d - \text{rank}\,[\nabla p_1(\boldsymbol{w}) \quad \cdots \quad \nabla p_K(\boldsymbol{w})]^\top$.

Although Definition 4.1 clearly indicates that the dimension represents the complexity of the set $V$, it is difficult to calculate the dimension according to this definition. Definition 4.2 shows that the dimension represents the algebraic complexity of the polynomial ring. Definition 4.3 characterizes the dimension based on the local structure of the affine variety, making it suitable for numerical calculation as discussed in Section 4.2. It generalizes the rank-nullity theorem $d_V = d - \text{rank}\,\boldsymbol{D}$ in the linear case, as mentioned in Section 3.3. The details of the concepts associated with these definitions are given in Appendix B.

### 4.1 Lower Bound

We demonstrate that the dimension $d_\mathcal{V}$ of the affine variety can be characterized by the linear part of the operator $\mathscr{D}$.

**Proposition 4.4.** *Suppose the operator $\mathscr{D}$ can be decomposed as $\mathscr{D} = \mathscr{L} + \mathscr{F}$, where $\mathscr{L}$ is a nonzero linear differential operator and $\mathscr{F}$ is a nonlinear operator. Then, we have $d_{\mathcal{V}(\mathscr{L})} \leq d_{\mathcal{V}(\mathscr{D})}$.*

Combining the result of Proposition 4.4 with Theorem 3.5 suggests that the nonlinear part $\mathscr{F}$ of the operator increases the affine variety dimension, which has a negative effect on generalization. Furthermore, the dimension of the affine variety associated with the linear part $\mathscr{L}$ can be easily calculated by the rank of the matrix. Therefore, the lower bound of the dimension of the affine variety associated with the nonlinear operator $\mathscr{D}$ can be easily determined, allowing us to estimate the minimum required amount of data $n$.

## 4.2 Numerical Calculation Method

According to Definition 4.2, the dimension of an affine variety is typically obtained by calculating the degree of the denominator of the Hilbert series, by using Gröbner bases. However, the worst-case time complexity of Buchberger's algorithm [7], the standard method for computing Gröbner bases, is double exponential in the number of variables $d$. Therefore, on the basis of 4.3, we approximate $d_{\mathcal{V}}$ by sampling $\boldsymbol{w}_1^*, \ldots, \boldsymbol{w}_N^*$ from the affine variety $\mathcal{V}$ with a suitable distribution and then computing $\max_{\boldsymbol{w}^* \in \{\boldsymbol{w}_1^*, \ldots, \boldsymbol{w}_N^*\}} d - \mathrm{rank}\left(\nabla^\top [p_1(\boldsymbol{w}^*), \ldots, p_K(\boldsymbol{w}^*)]^\top\right)$. When the operator $\mathscr{D}$ is nonlinear, we perform simulations with various boundary conditions and project the obtained solutions onto the basis $\mathcal{B}$ to sample $\boldsymbol{w}^* \in \mathcal{V}$. For linear operators, the dimension does not depend on the particular weight $\boldsymbol{w}$, and the rank of the matrix $\boldsymbol{D}$ discussed in Section 3.3 precisely determines $d_{\mathcal{V}}$. Assuming the use of standard rank computation algorithms, the computational complexity of this numerical approach is $\mathcal{O}(N \cdot \min(K, d)Kd)$ for the nonlinear case, and $\mathcal{O}(\min(K, d)Kd)$ for the linear case. This complexity is practical and feasible for most scenarios considered in our setting.

# 5 Experiments

To evaluate the generalization performance of physics-informed linear regression (PILR) compared to ridge regression (RR) using basis functions $\mathcal{B}$, we conducted experiments on representative differential equations. We varied the data size $n$ and parameter count $d$, and report test MSE (mean ± standard deviation) across 10 random initial or boundary conditions. Experimental details are provided in Appendix G.

When the operator $\mathscr{D}$ is linear, PILR approximates the solution to Eq. (3) as [16]:

$$\widehat{\boldsymbol{w}} = (\boldsymbol{\Phi}^\top \boldsymbol{\Phi} + n\boldsymbol{M})^{-1} \boldsymbol{\Phi}^\top \boldsymbol{y},$$

where $\boldsymbol{M}$ depends on hyperparameters $\xi$ and $\nu$ (see Eq. (6)); setting $\nu = 0$ yields RR.

For nonlinear equations, we train models by minimizing a soft-constrained loss using the Adam optimizer. Hyperparameters $\xi$ and $\nu$ are tuned via validation MSE.

## 5.1 Learning Strong Solutions

In this section, we investigate the strong solutions of the classical harmonic oscillator and the diffusion equation with periodic boundary conditions, by employing the Dirac measure, which corresponds to the collocation method used in PINNs. The solutions to these equations can be obtained analytically. Through these straightforward examples, we demonstrate both analytically and numerically that the generalization performance is determined by the dimension of the affine variety.

**Harmonic Oscillator**   The initial value problem of a harmonic oscillator $\mathscr{D}[y] = 0$ with a spring constant $k_s$ and mass $m_s$ in the domain $\Omega = [0, T]$ is given by:

$$\mathscr{D}[y] = \frac{\mathrm{d}^2}{\mathrm{d}t^2}y + \frac{k_s}{m_s}y, \quad y(0) = y_0, \quad \frac{\mathrm{d}}{\mathrm{d}t}y(0) = v_0, \tag{8}$$

where $y_0$ and $v_0$ are the initial position and velocity, respectively. The solution to the initial value problem is analytically given by $y(t) = y_0 \cos(\omega t) + \frac{v_0}{\omega}\sin(\omega t)$, $\omega = \sqrt{k_s/m_s}$. The settings for the basis and the trial functions with the measure $\phi_j \in \mathcal{B}$, $(\psi_k, \mu_k) \in \mathcal{T}$ of indices $1 \leq j \leq d_t$ and $1 \leq k \leq K_t$ are as follows:

$$\phi_1(x) = 1, \; \phi_{2j}(x) = \cos(\omega_j x), \; \phi_{2j+1}(x) = \sin(\omega_j x), \; \psi_k(x) = 1, \; \mu_k = \delta_{x_k}, \tag{9}$$

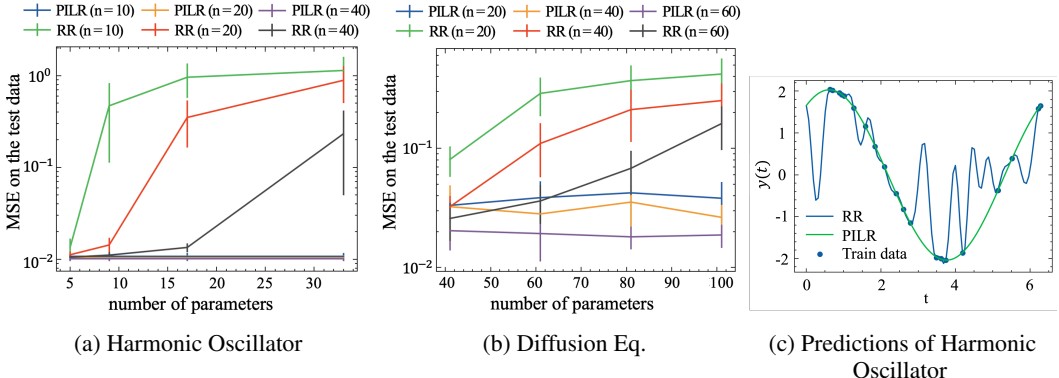

| (a) Harmonic Oscillator | (b) Diffusion Eq. | (c) Predictions of Harmonic Oscillator |

Figure 1: Experimental results for the strong solutions. (a, b) Test MSE (log scale) vs. number of parameters for the harmonic oscillator (a) and diffusion equation (b). The plots compare RR and PILR for three different data sizes $n$, showing the mean and standard deviation across 10 initializations. (c) Predictions of harmonic oscillator using a 33-parameter model trained on 20 samples: RR and PILR, with training data points indicated.

where $\omega_j := \frac{j\pi}{T}$ is the $j$-th frequency and $\delta_{x_k}$ is the Dirac measure centered at the point $x_k \in \Omega$, which is uniformly sampled from data.

Then, the dimension of the affine variety is $d_{\mathcal{V}} = 2$, representing the essential degrees of freedom of the solution. Figure 1a supports our theory experimentally. For RR, the generalization performance degrades as the number of parameters $d = 2d_t + 1$ increases due to overfitting, as shown in Fig. 1c. In contrast, for PILR, the performance remains stable regardless of the number of parameters $d$ owing to the lower dimension of the affine variety $d_{\mathcal{V}} = 2$.

**Diffusion Equation**  The initial value problem for the one-dimensional diffusion equation $\mathscr{D}[u] = 0$ with diffusion coefficient $c$ and periodic boundary conditions is given by:

$$\begin{cases} \frac{\partial u}{\partial t} - c\frac{\partial^2 u}{\partial x^2} = 0, & (x,t) \in [-\Xi, \Xi] \times [0, T] \\ u(x,0) = u_0(x), & x \in [-\Xi, \Xi] \\ u(-\Xi, t) = u(\Xi, t), \ \frac{\partial u}{\partial x}(-\Xi, t) = \frac{\partial u}{\partial x}(\Xi, t), & t \in [0, T] \end{cases} \tag{10}$$

We define the basis functions $\phi \in \mathcal{B}$ and the test functions with measures $(\psi, \mu) \in \mathcal{T}$ as follows:

$$\phi_{2j,j'} = \cos(\omega_j x)e^{-c\omega_{j'}^2 t}, \ \phi_{2j+1,j'} = \sin(\omega_j x)e^{-c\omega_{j'}^2 t}, \quad \psi_{k,k'} = 1, \ \mu_{k,k'} = \delta_{(t_k, x_{k'})}, \tag{11}$$

where the frequency is $\omega_j = j\pi/\Xi$. The indices are in the ranges $0 \leq j \leq d_x$, $0 \leq j' \leq d_t$, $1 \leq k \leq K_t$, and $1 \leq k' \leq K_x$.

The analytical solution is expressed as a linear combination of the above basis functions. The number of bases is $d = 2d_x d_t + 1$, while the dimension of an affine variety is given by $d_{\mathcal{V}} = 2\min(d_x, d_t) + 1$. Figure 1b shows the results when we set $\alpha = 1.0$, $j_{\max} = 1$, $d_t = 2$, and vary $d_x$. The results indicate that the generalization performance of PILR does not deteriorate as $d_x$ increases, in contrast to RR.

## 5.2 Learning Weak Solutions

In this section, we investigate weak solutions for the harmonic oscillator and the diffusion equation, employing a variational framework with Borel measures. The governing equations and basis functions are identical to those in Section 5.1.

**Harmonic Oscillator**  We define the trial functions $\psi_k$ for $1 \leq k \leq K_t$ as:

$$\psi_1(x) = 1, \quad \psi_{2k-1}(x) = \cos(\omega_k x), \quad \psi_{2k}(x) = \sin(\omega_k x), \tag{12}$$

where $\omega_k := \frac{k\pi}{T}$ is the frequency. The associated measure $\mu_k$ is the Lebesgue measure on $\Omega = [0, T]$.

The dimension of the affine variety remains $d_{\mathcal{V}} = 2$, consistent with the strong solutions. Experimental results in Fig. 2a confirm this. The performance of PILR is stable and independent of the number of basis functions, unlike RR, which shows performance degradation as model complexity increases.

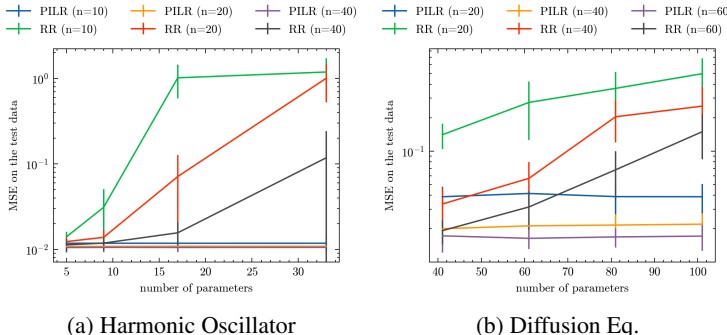

(a) Harmonic Oscillator              (b) Diffusion Eq.

Figure 2: Experimental results for the weak solutions. (a, b) Test MSE (log scale) vs. number of parameters for the harmonic oscillator (a) and diffusion equation (b). The plots compare RR and PILR for three different data sizes $n$, showing the mean and standard deviation across 10 initializations.

**Diffusion Equation**    The trial functions $\psi_{k,k'}$ combine a piecewise constant basis in time and a Fourier basis in space. For indices $1 \leq k \leq K_t$ and $1 \leq k' \leq K_x$, they are:

$$\psi_{k,2k'-1}(x,t) = 1_{[t_k,t_{k+1}]}(t)\cos(\omega_{k'}x), \quad \psi_{k,2k'}(x,t) = 1_{[t_k,t_{k+1}]}(t)\sin(\omega_{k'}x), \qquad (13)$$

where $1_{[t_k,t_{k+1}]}(t)$ is the indicator function for the time interval, defined as

$$1_{[t_k,t_{k+1}]}(t) = \begin{cases} 1 & \text{if } t \in [t_k, t_{k+1}) \\ 0 & \text{otherwise} \end{cases}, \qquad (14)$$

and $\omega_{k'} = \frac{k'\pi}{\Xi}$. The associated measure is the Lebesgue measure on $[-\Xi, \Xi] \times [0, T]$.

The dimension of the affine variety, $d_{\mathcal{V}} = 2\min(d_x, d_t) + 1$, is identical to the strong solution case. The results in Fig. 2b show that PILR's generalization performance remains robust as the number of spatial basis functions $d_x$ increases, demonstrating its advantage over RR.

### 5.3   Learning Numerical Solutions

In this section, we learn approximate solutions using numerical methods that use finite difference for four equations. In this setting, we consider the affine variety of the difference equation $\mathscr{D}_{\boldsymbol{h}}$ and the base functions $\mathcal{B}_{\boldsymbol{h}}$ and the trial functions with the measure $\mathcal{T}_{\boldsymbol{h}}$ corresponding to the numerical method with step size $\boldsymbol{h}$. We first validate our theory using linear and nonlinear Bernoulli equations discretized by the explicit Euler method.

**Discrete Bernoulli Equation**    We discretize the Bernoulli equation on the interval $\Omega = [0, T]$ with uniform step size $h$:

$$\mathscr{D}_h[y] \;=\; \frac{y_{\tau+1} - y_\tau}{h} \;+\; P\,y_\tau \;-\; Q\,y_\tau^\rho \;=\; 0, \quad \tau = 0, \ldots, n_t - 1, \quad n_t = \frac{T}{h},$$

where $y_\tau = y(t_\tau)$. We consider two parameter regimes $(P, Q, \rho)$ set to $(1.0, 0.0, 0.0)$ for the linear case and to $(1.0, 0.5, 2)$ for the non-linear case. The initial value $y_0$ is sampled from $\mathcal{N}(0, 1)$, and the reference solution is calculated explicitly by Euler. Further details on the choice of $n_t$, basis/trial functions, measure $(\psi_\tau, \mu_\tau)$, and implementation are given in Appendix G.2.

**Discrete Diffusion Equation**    We discretize the one-dimensional diffusion equation over $\Omega = [-\Xi, \Xi] \times [0, T]$ with step sizes $\boldsymbol{h} = (h_x, h_t)$ and diffusion coefficient $c(u)$:

$$\mathscr{D}_{\boldsymbol{h}}[u] = \frac{u_j^{\tau+1} - u_j^\tau}{h_t} - c(u_j^\tau)\frac{u_{j+1}^\tau - 2\,u_j^\tau + u_{j-1}^\tau}{h_x} \;=\; 0, \quad j = 1, \ldots, n_x,\ \tau = 1, \ldots, n_t,$$

where $u_j^\tau = u(x_j, t_\tau)$. We consider two cases: $c(u) = 1.0$ for the linear case and $c(u) = 0.1/(1+u^2)$ for the nonlinear case. Periodic boundary conditions are imposed in $x$. More details on the grid, the basis / trial functions, and the numerical setup are given in Appendix G.2.

Tables 1 and 2 show that PILR achieves a higher performance than RR for large values of $d$. While the dimension $d_{\mathcal{V}}$ is independent of the time discretization step size in the Euler method, it depends on the spatial discretization step size in the FDM. We include supplementary experiments in Appendix H, where we fix the ambient dimension $d$ and vary the size of the trial-function set $\mathcal{T}$.

Table 1: Experimental results for the discrete linear and nonlinear Bernoulli equations approximated by the explicit Euler method. The settings include various step sizes $h$. The number of parameters (basis) $d$, and the calculated dimension of the affine variety $d_\mathcal{V}$.

| Settings | $\mathscr{D}_h$ | Linear Bernoulli eq. | | Nonlinear Bernoulli eq. | |
|---|---|---|---|---|---|
| | $h$ | $1/100$ | $1/200$ | $1/100$ | $1/200$ |
| Dimensions | $d$ | 100 | 200 | 100 | 200 |
| | $d_\mathcal{V}$ | 1 | 1 | 1 | 1 |
| Test MSE | RR | $0.48 \pm 0.32$ | $0.63 \pm 0.43$ | $0.60 \pm 0.41$ | $0.72 \pm 0.49$ |
| | PILR | $0.012 \pm 0.0025$ | $0.011 \pm 0.0013$ | $0.013 \pm 0.0024$ | $0.013 \pm 0.0018$ |

Table 2: Experimental results for the discrete linear and nonlinear diffusion equations approximated by the FDM. The settings include various step sizes $\boldsymbol{h} = (h_t, h_x)$. The number of parameters (basis) $d$, and the calculated dimension of the affine variety $d_\mathcal{V}$.

| Settings | $\mathscr{D}_h$ | Linear diffusion eq. | | Nonlinear diffusion eq. | |
|---|---|---|---|---|---|
| | $(h_t, h_x)$ | $(1/400, 2/10)$ | $(1/400, 2/20)$ | $(1/200, 2/10)$ | $(1/200, 2/20)$ |
| Dimensions | $d$ | 4010 | 8020 | 2010 | 4020 |
| | $d_\mathcal{V}$ | 10 | 20 | 10 | 20 |
| Test MSE | RR | $2.21 \pm 0.56$ | $2.14 \pm 0.57$ | $1.12 \pm 0.40$ | $1.11 \pm 0.40$ |
| | PILR | $1.13 \pm 0.30$ | $0.79 \pm 0.16$ | $0.26 \pm 0.11$ | $0.22 \pm 0.10$ |

## 5.4   Impact of Basis Misspecification on Generalization

This section considers a practical scenario where the basis functions are misspecified, a situation that can occur during manual design or through random selection, as in an Extreme Learning Machine (ELM) [22]. Any such misspecification can degrade performance by increasing the **approximation error**. As detailed in Appendix C.4, the total error is composed of this approximation error and an estimation error. While our theory demonstrates that physical constraints can reduce the estimation error, the overall model performance is limited by the magnitude of the approximation error.

To demonstrate this effect, we conducted an experiment on the Harmonic Oscillator, intentionally omitting the known analytical frequency from the basis functions. Other experimental settings were identical to those in Section 5.1. With 10 data points, the performance was exceptionally poor. For a basis size of $d = 17$, the test MSE was approximately $1.435 \pm 0.646$, of which the approximation error constituted nearly the entire amount at $1.430$. Increasing the basis size to $d = 33$ had a negligible effect; the test MSE remained high at $1.434$ as the approximation error was unchanged.

This result clearly shows the total error being dominated by the approximation error. It underscores a prerequisite for our theory: the improvement in generalization from physics-informed constraints is achieved only when the model possesses sufficient expressive capacity to represent the true solution.

## 6   Conclusion

This study introduces a framework for analyzing physics-informed models through the lens of affine varieties induced by the governing differential equations. We establish that generalization performance is governed by the dimension of this variety, rather than the number of model parameters, a finding that unifies existing theories for linear equations. We further provide a method for calculating this dimension and present experimental validation confirming that this intrinsic dimension effectively mitigates overfitting in highly parameterized settings. Although our analysis centers on linear regression models, the proposed geometric framework is broadly applicable to both linear and nonlinear differential equations, as our experiments demonstrate. This work offers a foundational, geometric interpretation of generalization that establishes a promising, though challenging, direction for future theory-guided model selection, such as the optimal choice of basis and trial functions. Future work includes the extension and validation of our framework for other architectures, such as NN and ELM. The framework can also be extended to differential equations with unknown parameters by analyzing an augmented parameter space.

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

# A Notation

| Symbol | Description |
| --- | --- |
| **Data** | |
| $f^*$ | True function to be learned |
| $\Omega \subseteq \mathbb{R}^m$ | Input domain |
| $m$ | Input dimension |
| $n$ | Number of observations |
| $(x_i, y_i)$ | $i$-th observation ($x_i$: input, $y_i$: output) |
| $\boldsymbol{y}$ | Target vector $[y_1, \ldots, y_n]^\top$ |
| $\sigma^2$ | Noise variance |
| $\varepsilon_i$ | Normally distributed noise following $\mathcal{N}(0, \sigma^2)$ |
| **Affine Variety and Variables** | |
| $\mathscr{D}$ | Differential operator |
| $\mathcal{B} = \{\phi_j\}_{j=1}^d$ | Basis functions |
| $\phi_j$ | j-th basis function from $\mathcal{B}$ |
| $\boldsymbol{\phi}(x) \in \mathbb{R}^d$ | Basis vector at $x$ |
| $\boldsymbol{\Phi} \in \mathbb{R}^{n \times d}$ | Design matrix (i-th row is $\boldsymbol{\phi}(x_i)^\top$) |
| $\mathcal{T} = \{(\psi_k, \mu_k)\}_{k=1}^K$ | Finite collection of trial function and measure pairs. |
| $d$ | Number of basis functions $|\mathcal{B}|$ (ambient dimension) |
| $K$ | Number of trial functions $|\mathcal{T}|$ |
| $\mathcal{V}(\mathscr{D}, \mathcal{B}, \mathcal{T})$ | Affine variety defined by $\mathscr{D}, \mathcal{B}, \mathcal{T}$ (set of weight vectors) |
| $d_{\mathcal{V}}$ | Dimension of the affine variety $\mathcal{V}$ |
| $\boldsymbol{w}, \hat{\boldsymbol{w}}, \boldsymbol{w}^*$ | Weight vectors (learnable, estimated, optimal) |
| $\mathbb{B}_2(R)$ | $\ell_2$ ball of radius $R$ |
| $\mathcal{V}_R$ | Affine variety constrained by the $\ell_2$-ball ($\mathcal{V} \cap \mathbb{B}_2(R)$) |
| $\lambda_n$ | $L^2$ regularization parameter |
| $\|\cdot\|_2, \|\cdot\|$ | Vector $\ell_2$ norm, function $L^2$ norm w.r.t. Borel measure |
| $\langle\cdot,\cdot\rangle_2, \langle\cdot,\cdot\rangle_\mu$ | Vector Euclidean inner product, function inner product w.r.t. measure $\mu$ |
| **Geometric / Complexity Measures** | |
| $V \subseteq \mathbb{R}^d$ | General affine variety |
| $(\beta, d_V)$-regular set | Regularity condition |
| $d_V$ | Dimension of $V$ |
| $\text{codim}(L)$ | Codimension $d - d_L$ |
| $\beta$ | Upper bound on connected components of intersections |
| $\mathcal{N}(V, \varepsilon, \|\cdot\|_2)$ | $\varepsilon$-covering number w.r.t. $\ell_2$ norm $\|\cdot\|_2$. |
| **Analysis Constants** | |
| $M$ | Upper bound constant for the basis functions, such that $\|\boldsymbol{\phi}(x)\|_2 \leq M$ |
| $\eta$ | Upper bound constant for the lower eigenvalue of design matrix $\boldsymbol{\Phi}$. |
| $\Gamma$ | Stability constant of the estimator |
| $\delta$ | Probability parameter |
| **Linear Operators / PI Kernel** | |
| $\boldsymbol{D}, D_{k,j}$ | Constraint matrix when the operator $\mathscr{D}$ is linear; $D_{k,j}$ is its entry |
| $\mathscr{L}, \mathscr{F}$ | Linear, nonlinear parts of $\mathscr{D}$ |
| $d_{\mathcal{V}(\mathscr{L})}, d_{\mathcal{V}(\mathscr{D})}$ | Dimensions under $\mathscr{L}, \mathscr{D}$ |
| $\kappa_{\boldsymbol{M}}(x, y)$ | PI kernel defined with regularization matrix $\boldsymbol{M}$ |
| $\xi, \nu$ | Hyperparameters of the PI kernel (controlling the balance) |
| $\boldsymbol{B}, B_{j,j'}$ | Basis Gram matrix, its entry |
| $\boldsymbol{T}, T_{k,k'}$ | Trial Gram matrix, its entry |
| $d_{\text{eff}}(\xi, \nu)$ | Effective dimension of the PI kernel |
| $\sigma(\cdot), \alpha, \alpha_j$ | Spectrum of a matrix and its entry (eigenvalues) |
| **Dimension Calculation** | |
| $p_k$ | Defining polynomial |
| $N$ | Number of samples ($\boldsymbol{w}_1^*, \ldots, \boldsymbol{w}_N^*$) |

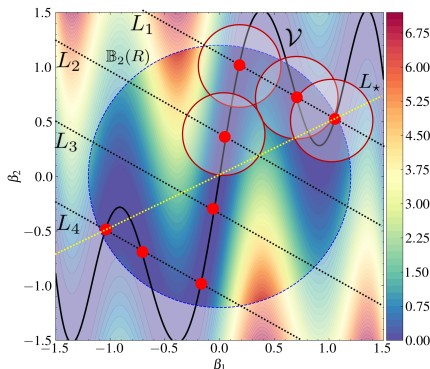

Figure 3: Illustration of the construction of the $\varepsilon$-covering of the affine variety $\mathcal{V} \subseteq \mathbb{R}^2$ and the associated loss landscape. The black curve represents a $(\beta, d_{\mathcal{V}})$ regular affine variety with dimension $d_{\mathcal{V}} = 1$. The color gradients depict the loss landscape $\mathcal{L}(\boldsymbol{w}) := \sum_{k=1}^{K} \|p_k(\boldsymbol{w})\|_2^2$ of the equations defining $\mathcal{V} = \{\boldsymbol{w} : p_k(\boldsymbol{w}) = 0, \; \forall k = 1, \ldots, K\}$. The blue dotted line represents a $\ell_2$ ball of radius $R$. The affine variety constrained with the $\ell_2$ ball is covered by $\varepsilon$-balls centered at the intersections of $\mathcal{V}$ with four given subspaces $\{L_s\}_{s=1}^4$, shown as red points. The upper bound on the number of intersections of every subspace with the variety is $\beta$, while the actual maximum number is 5 formed by the subspace $L_\star$ (the yellow dotted line). The loss landscape of the equations is zero on $\mathcal{V}$ and locally convex around the points in $\mathcal{V}$.

# B  Mathematical Background on Affine Varieties

In this section, we provide a formal definition of several concepts related to affine varieties and review the definition of the dimension of an affine variety, as briefly described in Section 4.

An affine variety is a fundamental concept in algebraic geometry. It is a subset of an affine space, defined as the solution set to a system of polynomial equations. Let $\mathbb{K}[\boldsymbol{w}]$ denote the set of polynomials in the variables $\boldsymbol{w} = (w_1, \ldots, w_d) \in \mathbb{K}^d$ over a field $\mathbb{K}$ (often $\mathbb{R}$ or $\mathbb{C}$). An affine variety $V(p_1, \ldots, p_K) \subseteq \mathbb{K}^d$ defined by the polynomials $p_1, \ldots, p_K \in \mathbb{K}[\boldsymbol{w}]$ is given by:

$$V(p_1, \ldots, p_K) := \left\{ \boldsymbol{w} \in \mathbb{K}^d : p_k(\boldsymbol{w}) = 0, \; \forall k = 1, \ldots, K \right\}.$$

The geometry of an affine variety is determined by the set of all polynomials that "vanish" on $V$, i.e., those that become zero for every point in $V$. This set is called the ideal of the affine variety, denoted $I(V)$, and is defined as follows:

$$I(V) := \{p \in \mathbb{K}[\boldsymbol{w}] : p(\boldsymbol{w}) = 0, \; \forall \boldsymbol{w} \in V\}.$$

The generating polynomial set $\{p_k\}_{k=1}^K$ of the affine variety $V$ is a subset of the ideal $I(V)$.

The coordinate ring over $V$, denoted $\mathbb{K}[V]$, is introduced to identify polynomials that yield the same values on the variety $V$. Specifically, $\mathbb{K}[V]$ is defined as the quotient of the polynomial ring $\mathbb{K}[\boldsymbol{w}]$ by the ideal $I(V)$, i.e., $\mathbb{K}[\boldsymbol{w}]/I(V)$. In the coordinate ring $\mathbb{K}[V] = \mathbb{K}[\boldsymbol{w}]/I(V)$, the difference between $p$ and $q$ vanishes on $V$, i.e., $p(\boldsymbol{w}) = q(\boldsymbol{w})$ for all $\boldsymbol{w} \in V$, or equivalently $p - q \in I(V)$. Thus, $p$ and $q$ are considered the same element. From another viewpoint, the coordinate ring $\mathbb{K}[V]$ can be considered as a set of polynomials not included in the ideal $I(V)$.

Based on the above definitions, we review the definition of the dimension $d_V$ of the affine variety in Appendix B.1 and the regularity in Appendix B.2.

## B.1  Dimension of Affine Varieties

### B.1.1  Geometric View

Considering the affine variety $V$ as an affine space, we can naturally define a subvariety as an "subset" of the variety that also satisfies polynomial equations. Let $q_1, \ldots, q_S$ be polynomials in a ring. Define $\langle q_1, \ldots, q_S \rangle$ as the smallest ideal generated by $q_1, \ldots, q_S$; that is, $\langle q_1, \ldots, q_S \rangle$ consists of all finite

sums of the form $\sum_{i=1}^{S} r_i q_i$ where each $r_i$ is in the ring: $\langle q_1, \ldots, q_S \rangle = \{\sum_{i=1}^{S} r_i q_i\}$. A subvariety $U$ of $V$ is defined as the zero set of a subset ideal $\langle q_1, \ldots, q_S \rangle \subseteq \mathbb{K}[\boldsymbol{w}]/I(V)$ given by:

$$U := \left\{ \boldsymbol{w} \in \mathbb{K}^d : q_s(\boldsymbol{w}) = 0, \ \forall q_s \in \langle q_1, \ldots, q_S \rangle \right\}.$$

By using the concept of subvarieties, the dimension of an affine variety is defined as follows:

**Definition 4.1.** The maximal length of the chains $V_0 \subset V_1 \subset \ldots \subset V_{d_V}$ of non-empty subvarieties of $V$.

This definition intuitively represents the size of $V$ by the maximal length of an increasing sequence of subspaces. If the generating polynomials $\{p_k\}_{k=1}^{K}$ are all linear, the dimension of $V$ is defined as the maximal length of an increasing sequence of linear subspaces within $V$, which corresponds to the dimension of $V$ as a linear space.

When we focus on the local structure, the following equivalent definition is obtained:

**Definition 4.3.** The maximal dimension of the tangent vector spaces at the non-singular points $U \subseteq V \subset \mathbb{R}^d$ of the variety, *i.e.*, $d_V = \max_{\boldsymbol{w} \in U} d - \mathrm{rank} \left[ \nabla p_1(\boldsymbol{w}) \quad \cdots \quad \nabla p_K(\boldsymbol{w}) \right]^\top$.

From this definition, we can see that the dimension $d_V$ is a global quantity that summarizes the local linearized structure of the affine variety $V$ at a point.

For example, let $\mathbb{K} = \mathbb{R}$ and $V \subset \mathbb{R}^3$ be the plane: $V = \{(x, y, z) : x + y - z = 0\}$. A chain of subvarieties within $V$ is $V_0 \subset V_1 \subset V_2$, where $V_0 = \{(0, 0, 0)\}$ (a point, 0-dimensional), $V_1 = \{(t, 0, t) : t \in \mathbb{R}\}$ (a line, 1-dimensional), and $V_2 = V$ itself (the plane, 2-dimensional). The maximal length of the nested subvarieties is two, *i.e.*, $d_V = 2$, which means that a plane has two degrees of freedom.

### B.1.2 Algebraic View

The structure of an affine variety is determined by the ideal $I(V)$. Intuitively, the larger $I(V)$ is, the more polynomial constraints there are, which means that $V$ becomes smaller, and consequently, the coordinate ring $\mathbb{K}[V]$ also becomes smaller. From this perspective, it is natural to expect a deep connection between the dimension of the coordinate ring $\mathbb{K}[V]$ (and similarly the ideal $I(V)$) and the dimension of the affine variety $V$.

To explore this connection, we first discuss the dimension of the coordinate ring $\mathbb{K}[V]$ using Krull dimension. The ideal $\mathfrak{p} \subset \mathcal{R}$ in a polynomial ring $\mathcal{R}$ is prime if $\forall a, b \in \mathcal{R}, \ ab \in \mathfrak{p} \Rightarrow a \in \mathfrak{p}$ or $b \in \mathfrak{p}$. The definition of the dimension of the affine variety through the Krull dimension is shown below.

**Definition B.1.** The Krull dimension of the coordinate ring $\mathbb{K}[V]$: The maximum length $d$ of the chain of prime ideals $\mathfrak{p}_0 \subset \mathfrak{p}_1 \subset \cdots \subset \mathfrak{p}_d$ in the coordinate ring $\mathbb{K}[V]$.

This definition signifies that the dimension of an affine variety is characterized in the world of polynomial sets by the maximal length of an increasing chain of "subsets" within the coordinate ring, corresponding to Definition 4.1 from a geometric perspective.

In contrast, the size of the coordinate ring $\mathbb{K}[V]$ can also be measured using Hilbert series. First, by homogenizing the defining equations by adding one variable $\gamma \in \mathbb{K}$, we embed the affine variety $V \subset \mathbb{K}^d$ into the projective variety $\mathcal{P} \subset \mathbb{K}^{d+1}$. The projective variety $\mathcal{P}(h_1, \ldots, h_K) \subset \mathbb{K}^{d+1}$, defined by the homogeneous polynomials $h_1, \ldots, h_K \in \mathbb{K}[(\boldsymbol{w}, \gamma)]$, is given by:

$$\mathcal{P}(h_1, \ldots, h_K) := \left\{ (\boldsymbol{w}, \gamma) \in \mathbb{K}^{d+1} : h_k(\boldsymbol{w}, \gamma) = 0, \ \forall k = 1, \ \ldots, \ K \right\}.$$

The dimension of the variety is also increased by one, i.e., $d_{\mathcal{P}} = d_V + 1$. The coordinate ring $\mathbb{K}[\mathcal{P}] = \mathbb{K}[(\boldsymbol{w}, \gamma)]/I(\mathcal{P})$ of the projective variety $\mathcal{P}$ can be decomposed into subgroups (called the graded coordinate ring) as follows:

$$\mathbb{K}[\mathcal{P}] = \bigoplus_{\rho \in \mathbb{N}} S_\rho, \ S_0 = \mathbb{K},$$

where $S_\rho$ is the set of homogeneous polynomials of degree $\rho$ modulo the ideal $I(\mathcal{P})$. As a metric for the size of the coordinate ring $\mathbb{K}[\mathcal{P}]$, the Hilbert function $\mathrm{H}(\rho)$ and Hilbert-Poincaré series $\mathrm{HS}(t)$ are

defined as follows:

$$H(\rho) = \dim S_\rho, \ HS(t) = \sum_{\rho \in \mathbb{N}} H(\rho) t^\rho = \frac{\prod_{k=1}^{K}(1 - t^{\rho_k})}{(1-t)^{d+1}},$$

where $\dim$ denotes the Krull dimension and $\rho_1, \ldots, \rho_K$ are the degrees of the homogeneous polynomials $h_1, \ldots, h_K$.

The Hilbert function represents the dimension of a "subspace" of the decomposed coordinate ring, and the Hilbert series is the generating function of the sequence of the Hilbert function, which is also a rational function with a pole at $t = 1$. These measures indicate the growth of the dimension of the homogeneous components of the algebra with respect to the degree. According to the dimension theorem, the Krull dimension of the projective variety $\mathcal{P}$ matches the order of the Hilbert series at the pole $t = 1$, which is one of the most important results in commutative algebra.

Therefore, the dimension of the affine variety is defined using the Hilbert series, as follows:

**Definition 4.2.** The degree of the denominator of the Hilbert series of the affine variety $V$.

Given the Gröbner basis of the ideal $I(\mathcal{P})$, the Hilbert series can be easily computed, leading to an efficient estimation of the dimension of the affine variety $d_V$.

## B.2 Regularity of Affine Varieties

We informally define the concept of a regular set for real affine varieties, which is used in Section 3.2 (for a formal definition, see Definition 2.1 in [42]).

A affine variety $V \subseteq \mathbb{R}^d$ is a $(\beta, d_V)$-regular set if:

1. For almost all affine planes $L$ with $\mathrm{codim}(L) \leq d_V$ in $\mathbb{R}^d$, $V \cap L$ has at most $\beta$ path-connected components.
2. For almost all affine planes $L$ with $\mathrm{codim}(L) > d_V$ in $\mathbb{R}^d$, $V \cap L$ is empty.

The notion $\mathrm{codim}$ represents the *codimension*. For an affine subspace $L \subseteq \mathbb{R}^d$, its codimension is defined by $\mathrm{codim}(L) = d - d_L$. Simply put, codimension is how many dimensions you are "missing" when comparing a smaller space inside a bigger space.

A regular set restricts the complexity of a variety $V$. Intuitively, the complexity of $V$ can be measured by the number of connected components in its cross sections. For instance, a complex shape may have cross sections that split into multiple connected components. The larger the number of connected components $\beta$, the more complex the topology of $V$. Moreover, the dimension at which we slice the variety is also important. If the slice (affine plane) is large enough in dimension, *i.e.*, the codimension is small ($< d_V$), then any intersection of the slice with $V$ is limited to at most $\beta$ connected pieces. Otherwise, the slice typically does not intersect $V$ at all. For example, consider the circle $V = \{(x,y) \in \mathbb{R}^2 : x^2 + y^2 - 1 = 0\}$. A line ($\mathrm{codim}(L) = 1$) intersects the circle in at most two points. For a single point ($\mathrm{codim}(L) = 2$), almost all points do not lie in the circle; that is, intersections with higher codimension affine subspaces are almost empty. This implies that the circle is a $(2, 1)$-regular set.

## B.3 Covering number of Affine Varieties

**Lemma B.2** (Zhang and Kileel [42]). *Let $V \subset \mathbb{R}^d$ be a $(\beta, d_V)$-regular set in the ball $\mathbb{B}_2(R)$ with the radius $R$. Then for all $\varepsilon \in (0, \mathrm{diam}(V)]$,*

$$\log \mathcal{N}(V, \varepsilon, \|\cdot\|_2) \leq d_V \log\left(\frac{2R d_V d}{\varepsilon}\right) + \log 2\beta. \tag{15}$$

This upper bound is obtained by slicing the affine variety $V$ with subspaces $\{L_s\}_{s \in \mathbb{N}}$ within $\mathbb{R}^d$ and covering $V$ with balls centered at the intersections of $L_s$ and $V$, *i.e.*, $V \subset \bigcup_s \bigcup_{v \in V \cap L_s} \mathbb{B}_2(v; \varepsilon)$.

The covering for the two-dimensional case is illustrated in Fig. 3. The first term, $(2R d_V d/\varepsilon)^{d_V}$, represents the number of subspaces $L_s$ needed to cover the entire space. It is mainly determined by the intrinsic dimension $d_V$ of the affine variety, although it is still influenced by the ambient

dimension $d$. The quantity $\beta$ in the second term denotes the number of intersections between a single subspace $L$ and the variety $V$, and represents the covering number of $V \cap L$. Topologically, it corresponds to the Betti numbers of the affine variety, which informally represent the number of holes in $V$. The upper bound on the quantity $\beta$ is given, for example, by the Petrovskii-Oleinik-Milnor inequality [33, 32, 28]. Specifically, an affine variety $V \cap \mathbb{B}_2(R)$ defined by polynomials $\{p_k\}_{k \in [K]}$ of maximum degree $\rho$ and the $\ell_2$-ball is $(\rho(2\rho - 1)^{d+1}, d_V)$-regular. This intuitively suggests that as the maximum degree of polynomials increases, the topology of the affine variety becomes more complex.

## C  Detailed Background on Problem Formulation

This appendix expands the formulation introduced in the main text, highlighting why a *hybrid* physics–data approach is required.

### C.1  Governing System

Let $\Omega \subset \mathbb{R}^d$ be a bounded domain with boundary $\partial\Omega$. For a differential operator $\mathscr{D} \colon L^2(\Omega) \to L^2(\Omega)$, the *true* state $f^* \colon \Omega \to \mathbb{R}$ satisfies the boundary-value problem

$$\mathscr{D}[f^*] = v \qquad \text{in } \Omega, \tag{16}$$
$$f^* = g \qquad \text{on } \partial\Omega, \tag{17}$$

where $v$ and $g$ are smooth but may be only *partially observed* or inferred indirectly.

### C.2  Available Information

In practice one seldom knows $v$ and $g$ exactly; instead one has:

- **Noisy pointwise observations.** A dataset $\{(x_i, y_i)\}_{i=1}^{N_u}$ with $y_i = f^*(x_i) + \varepsilon_i,\ \varepsilon_i \sim \mathcal{N}(0, \sigma^2)$, where $x_i \in \Omega \cup \partial\Omega$.
- **Weak-form physics information.** Linear functionals $l_k(u) \coloneqq \langle u, \psi_k \rangle_{\mu_k},\ k = 1, \ldots, N_r$, with trial functions $\psi_k \in C_c^\infty(\Omega)$ and measures $\mu_k$, together with the corresponding targets $l_k(v) = l_k(\mathscr{D}[f^*])$.

### C.3  Hybrid Surrogate Model

A representative hybrid approach is the *Physics-Informed Neural Network* (PINN) [36]. Given a neural surrogate $f_{\boldsymbol{w}} \colon \Omega \to \mathbb{R}$, its parameters $\boldsymbol{w}$ are obtained by minimising

$$\mathcal{L}(\boldsymbol{w}) = \underbrace{\frac{1}{N_u} \sum_{i=1}^{N_u} |f_{\boldsymbol{w}}(x_i) - y_i|^2}_{\text{data fidelity}} + \lambda \underbrace{\frac{1}{N_r} \sum_{k=1}^{N_r} |l_k(\mathscr{D}[f_{\boldsymbol{w}}]) - l_k(v)|^2}_{\text{physics residual (weak form)}}, \tag{18}$$

with hyper-parameter $\lambda > 0$ balancing empirical fit and physical consistency.

**Connection to standard collocation method**  In the standard collocation method, the unified residual forms reduce to pointwise strong-form residuals: specifically, for each $k$, we set

$$\psi_k(x) = 1, \quad \mu_k = \delta_{x_k},$$

where $\delta_{x_k}$ is the Dirac measure at collocation point $x_k$. In this case, the linear functional $l_k$ becomes

$$l_k(u) = \langle u, \psi_k \rangle_{\mu_k} = \int u(x)\, \mathrm{d}\delta_{x_k} = u(x_k),$$

and thus the physics term penalizes the pointwise physics residuals:

$$\sum_{k=1}^{N_r} |\mathscr{D}[f_{\boldsymbol{w}}](x_k) - v(x_k)|^2.$$

**Limiting regimes.**

- *Pure data fitting:* $\lambda = 0$ reduces Eq. (18) to standard supervised learning on $\mathcal{D}_u$.
- *Fully physics-informed:* If $\{l_k\}$ is dense and $N_r \to \infty$, the residual term enforces Eq. (16) everywhere.
- *Truly hybrid:* Finite $N_r$ with incomplete $\{l_k\}$—typical in engineering—captures partial physics, while the data term compensates for the missing information.

## C.4 Error decomposition and analysis

To directly measure how physics-based inductive bias improves the generalization ability of the machine learning models, we fix the physics information as a known immutable prior. Let $\mathcal{H}_{\text{base}}$ denote the unrestricted hypothesis class (e.g. linear models or neural networks). We consider two ways of incorporating the physics prior into the base hypothesis class $\mathcal{H}_{\text{base}}$:

- **Hard setting:** Enforce the differential equation residual exactly by:

$$\hat{f}_{\text{hard}} \in \underset{f \in \mathcal{H}_{\text{base}}}{\arg\min} \sum_{k=1}^{N_r} |l_k(\mathscr{D}[f]) - l_k(v)|^2 := \mathcal{H}_{\text{hard}}. \tag{19}$$

- **Soft setting:** Allow a relaxed residual tolerance:

$$\hat{f}_{\text{soft}}(\varepsilon) \in \left\{ f \in \mathcal{H}_{\text{base}} : \sum_{k=1}^{N_r} |l_k(\mathscr{D}[f]) - l_k(v)|^2 \leq \varepsilon \right\} := \mathcal{H}_{\text{soft}}. \tag{20}$$

For $\mathcal{H} \in \{\mathcal{H}_{\text{hard}}, \mathcal{H}_{\text{soft}}(\varepsilon)\}$, let $\hat{f} \in \mathcal{H}$ be the learned solution and $f_{\mathcal{H}}^* := \arg\min_{f \in \mathcal{H}} \|f^* - f\|$ the best attainable approximation. By the triangle inequality, we have:

$$\|f^* - \hat{f}\| \leq \|f^* - f_{\mathcal{H}}^*\| + \|f_{\mathcal{H}}^* - \hat{f}\|. \tag{21}$$

In this decomposition:

- The term $\|f^* - f_{\mathcal{H}}^*\|$ represents the approximation error: the best achievable error within the hypothesis class $\mathcal{H}$.
- The term $\|f_{\mathcal{H}}^* - \hat{f}\|$ represents the estimation error: the deviation due to finite data.

Our primary focus is to derive bounds on the estimation error $\|f_{\mathcal{H}}^* - \hat{f}\|$, thereby quantifying how well the learned solution converges to the best physics-constrained approximation.

In particular, our analysis centers on the hard constraint setting, where $\mathcal{H} = \mathcal{H}_{\text{hard}}$ on the linear base hypothesis

$$\mathcal{H}_{\text{base}} = \left\{ f_{\boldsymbol{w}} = \boldsymbol{w}^\top \boldsymbol{\phi} : \boldsymbol{w} \in \mathbb{R}^d, \ \phi_j \in \mathcal{B} \right\},$$

with $\mathcal{B}$ a chosen basis. Under the hard constraint, the admissible hypothesis class amounts to restricting the parameter vector $\boldsymbol{w}$ to lie on an affine variety induced by the PDE residuals. More explicitly,

$$\mathcal{H}_{\text{hard}} = \left\{ f_{\boldsymbol{w}} : \boldsymbol{w} \in \mathcal{V}(\mathscr{D}, \mathcal{B}, \mathcal{T}), \ \phi_j \in \mathcal{B} \right\},$$

where $\mathcal{V}(\mathscr{D}, \mathcal{B}, \mathcal{T})$ denotes the set of coefficient vectors $\boldsymbol{w}$ satisfying the algebraic constraints generated by the operator $\mathscr{D}$ acting on the basis $\mathcal{B}$ and tested against the functionals $\mathcal{T} = \{l_k\}_{k=1}^{N_r}$.

## C.5 Extension: Incomplete Operators with Learnable Parameters

The above framework can be generalized to settings where the governing differential operator itself is only partially known and contains learnable parameters. Formally, suppose that instead of a fixed operator $\mathscr{D} : L^2(\Omega) \to L^2(\Omega)$, we consider a parametric operator

$$\mathscr{D}_{\boldsymbol{c}} : L^2(\Omega) \to L^2(\Omega), \qquad \boldsymbol{c} \in \mathbb{R}^m,$$

where $\boldsymbol{c}$ denotes a vector of unknown coefficients to be simultaneously estimated from data. In this case, the admissible hypothesis space is naturally

$$\mathcal{H}_{\text{aug}} := \left\{ (f, \boldsymbol{c}) : f \in \mathcal{H}_{\text{base}}, \ l_k(\mathscr{D}_{\boldsymbol{c}}[f]) = l_k(v), \ k = 1, \ldots, N_r \right\},$$

defined as an *augmented constraint set* over the joint variable $(f, \boldsymbol{c})$.

The error decomposition then applies in this extended space: for $(f^*, \boldsymbol{c}^*)$ denoting the best attainable pair in $\mathcal{H}_{\text{aug}}$, the learned solution $(\hat{f}, \hat{\boldsymbol{c}})$ satisfies

$$\| f^* - \hat{f} \| \ \leq \ \underbrace{\| f^* - f^*_{\mathcal{H}_{\text{aug}}} \|}_{\text{approximation error}} + \underbrace{\| f^*_{\mathcal{H}_{\text{aug}}} - \hat{f} \|}_{\text{estimation error}},$$

with both terms now understood relative to the augmented parameter space.

**Typical cases.**

- **Unknown diffusion coefficient.** Consider the diffusion equation $\partial_t u - c\Delta u = 0$, where the diffusion constant $c > 0$ is unknown. Here $\boldsymbol{c} = (c)$ is a scalar parameter. Under the hard constraint, the admissible hypothesis class can be written as

$$\mathcal{H}_{\text{aug}} = \left\{ (f_{\boldsymbol{w}}, c) : (\boldsymbol{w}, c) \in \mathcal{V}\big(\partial_t f_{\boldsymbol{w}} - c\Delta f_{\boldsymbol{w}}, \mathcal{B}, \mathcal{T}\big), \ \phi_j \in \mathcal{B} \right\},$$

  where $\mathcal{V}(\cdot)$ denotes the algebraic variety of coefficient–parameter pairs $(\boldsymbol{w}, c)$ that satisfy the residual constraints induced by $\mathcal{T} = \{l_k\}_{k=1}^{N_r}$. Thus both approximation and estimation errors are quantified in this augmented parameter space.

- **Unknown diffusion term (learned surrogate).** In cases where the diffusion operator itself is not specified, one may introduce a surrogate $v_\theta$ to represent its action. The PDE constraint becomes

$$\partial_t f_{\boldsymbol{w}} - v_\theta = 0,$$

  leading to the hypothesis class

$$\mathcal{H}_{\text{aug}} = \left\{ (f_{\boldsymbol{w}}, v_\theta) : (\boldsymbol{w}, \theta) \in \mathcal{V}\big(\partial_t f_{\boldsymbol{w}} - v_\theta, \mathcal{B}, \mathcal{T}\big), \ \phi_j \in \mathcal{B} \right\}.$$

  Here $v_\theta$ serves as a learnable proxy for the unknown diffusion term. Our error decomposition applies verbatim in this augmented parameter space, with approximation error defined relative to the best attainable pair $(\boldsymbol{w}^*, \theta^*)$ and estimation error measuring the deviation of the learned $(\hat{\boldsymbol{w}}, \hat{\theta})$ from this target.

In summary, by enlarging the hypothesis space to include both explicit unknown coefficients and implicit unknown operator surrogates, the proposed error decomposition continues to hold, thereby providing a principled means of quantifying generalization in operator-learning settings with incomplete physics.

# D  Proof for Theorem 3.5

**Theorem 3.5** (Minimax Risk Bound). *Let $\mathcal{V}(\mathscr{D}, \mathcal{B}, \mathcal{T})$ be the $(\beta, d_\mathcal{V})$-regular affine variety defined in Eq. (2). Suppose Assumptions 3.2-3.4 hold. Then, there exists a positive constant $C$, independent of $n$, $d_\mathcal{V}$, $d$, and $\beta$, such that for any $\delta \in (0, 1)$, with probability at least $1 - \delta$, the minimax risk for PILR defined by Eq. (4) is bounded by*

$$\min_{\hat{\boldsymbol{w}}} \max_{\boldsymbol{w}^* \in \mathcal{V}_R} \| \hat{\boldsymbol{w}} - \boldsymbol{w}^* \|_2^2 \leq C\eta^{-1}\sigma M\Gamma R \left( \sqrt{\frac{d_\mathcal{V} \log(d_\mathcal{V} d)}{n}} + \sqrt{\frac{\log 2\beta}{n}} + 2\sqrt{\frac{\log(2/\delta)}{n}} \right). \quad (5)$$

*Proof.* **Step 1:** We first upper bound the prediction error by a term that represents the supremum of a empirical process in the metric space of the affine variety. Using Lemma D.1, we get:

$$\| \boldsymbol{\Phi}(\boldsymbol{w}^* - \hat{\boldsymbol{w}}) \|_2^2 \leq 2\boldsymbol{\varepsilon}^\top \boldsymbol{\Phi}(\boldsymbol{w}^* - \hat{\boldsymbol{w}}).$$

We denote $\mathrm{x}_{\boldsymbol{w}} := \boldsymbol{\varepsilon}^\top \boldsymbol{\Phi}(\boldsymbol{w} - \hat{\boldsymbol{w}})$ as the random process in the metric space $(\mathcal{V}_R, \|\cdot\|_2)$. Note that the estimator $\hat{\boldsymbol{w}}$ is a random variable depending on the parameter $\boldsymbol{w}$ and the noise $\boldsymbol{\varepsilon}$. Then, the minimax risk is bounded as follows.

$$\min_{\hat{\boldsymbol{w}}} \max_{\boldsymbol{w}^* \in \mathcal{V}_R} \|\hat{\boldsymbol{w}} - \boldsymbol{w}^*\|_2^2 \leq \min_{\hat{\boldsymbol{w}}} \max_{\boldsymbol{w}^* \in \mathcal{V}_R} \frac{\eta^{-1}}{n} \|\boldsymbol{\Phi}(\hat{\boldsymbol{w}} - \boldsymbol{w}^*)\|_2^2 \leq \frac{2}{n} \eta^{-1} \sup_{\boldsymbol{w} \in \mathcal{V}_R} \mathrm{x}_{\boldsymbol{w}}. \tag{22}$$

The first inequality holds by Assumption 3.3.

**Step 2:** Next, we calculate the supremum of the empirical process $\mathrm{x}_{\boldsymbol{w}}$ using the covering number. For all $\boldsymbol{w}_1, \boldsymbol{w}_2 \in \mathcal{V}_R$, it is shown that the variable $\mathrm{x}_{\boldsymbol{w}_1} - \mathrm{x}_{\boldsymbol{w}_2}$ has sub-Gaussian increments with respect to the metric $\|\cdot\|_2$:

$$\begin{aligned}
\mathrm{x}_{\boldsymbol{w}_1} - \mathrm{x}_{\boldsymbol{w}_2} &= \sum_{i=1}^{n} \varepsilon_i ((\boldsymbol{w}_1 - \hat{\boldsymbol{w}}_1) - (\boldsymbol{w}_2 - \hat{\boldsymbol{w}}_2))^\top \boldsymbol{\phi}(x_i) \\
&\leq \sum_{i=1}^{n} \varepsilon_i \|(\boldsymbol{w}_1 - \boldsymbol{w}_2) - (\hat{\boldsymbol{w}}_1 - \hat{\boldsymbol{w}}_2)\|_2 \|\boldsymbol{\phi}(x_i)\|_2 \\
&\leq \sum_{i=1}^{n} \varepsilon_i (\|\boldsymbol{w}_1 - \boldsymbol{w}_2\|_2 + \|\hat{\boldsymbol{w}}_1 - \hat{\boldsymbol{w}}_2\|_2) M \\
&\leq \Gamma \|\boldsymbol{w}_1 - \boldsymbol{w}_2\|_2 M \mathrm{e},
\end{aligned} \tag{23}$$

where $\mathrm{e}$ is the zero-mean Gaussian random variable with variance $n\sigma^2$. The second inequality holds by the Cauchy-Schwarz inequality and the third holds by the triangle inequality and Assumption 3.2. The last inequality holds by Assumption 3.4.

From Eq. (23), the random process $\mathrm{x}_{\boldsymbol{w}_1} - \mathrm{x}_{\boldsymbol{w}_2}$ has sub-Gaussian increments as follows.

$$\|\mathrm{x}_{\boldsymbol{w}_1} - \mathrm{x}_{\boldsymbol{w}_2}\|_{\psi_2} \leq \sqrt{n} \sigma M \Gamma \|Z\|_{\psi_2} \|\boldsymbol{w}_1 - \boldsymbol{w}_2\|_2,$$

where $Z$ is the standard Gaussian random variable and $\|\cdot\|_{\psi_2}$ is the sub-Gaussian norm. For the centered random process $\mathrm{z}_{\boldsymbol{w}} := \mathrm{x}_{\boldsymbol{w}} - \mathbb{E}[\mathrm{x}_{\boldsymbol{w}}]$, $\|\mathrm{z}_{\boldsymbol{w}_1} - \mathrm{z}_{\boldsymbol{w}_2}\|_{\psi_2} \lesssim \|\mathrm{x}_{\boldsymbol{w}_1} - \mathrm{x}_{\boldsymbol{w}_2}\|_{\psi_2}$ holds because $\|\mathrm{x}_{\boldsymbol{w}_1} - \mathrm{x}_{\boldsymbol{w}_2}\|_{\psi_2}$ is sub-Gaussian.

Using Lemma D.2, we obtain the following bound with some constant $C_0$:

$$\mathbb{E} \sup_{\boldsymbol{w} \in \mathcal{V}_R} \mathrm{z}_{\boldsymbol{w}} \leq C_0 \sqrt{n} \sigma M \Gamma R \left( \sqrt{d_\mathcal{V} \log d_\mathcal{V} d} + \sqrt{\log 2\beta} \right). \tag{24}$$

Next, using Dudley's integral tail bound, we have:

$$\Pr \left( \sup_{\boldsymbol{w} \in \mathcal{V}_R} \mathrm{z}_{\boldsymbol{w}} \leq \mathbb{E} \sup_{\boldsymbol{w} \in \mathcal{V}_R} \mathrm{z}_{\boldsymbol{w}} + C_0 \sqrt{n} \sigma M \Gamma 2R \sqrt{\log(\delta/2)} \right) \geq 1 - \delta.$$

By incorporating the non-centered process $\mathrm{x}_{\boldsymbol{w}}$, we obtain:

$$\Pr \left( \sup_{\boldsymbol{w} \in \mathcal{V}_R} \mathrm{x}_{\boldsymbol{w}} \leq \sup_{\boldsymbol{w} \in \mathcal{V}_R} |\mathbb{E}[\mathrm{x}_{\boldsymbol{w}}]| + \mathbb{E} \sup_{\boldsymbol{w} \in \mathcal{V}_R} \mathrm{z}_{\boldsymbol{w}} + C_0 \sqrt{n} \sigma M \Gamma 2R \sqrt{\log(\delta/2)} \right) \geq 1 - \delta. \tag{25}$$

To bound $\mathbb{E}[\mathrm{x}_{\boldsymbol{w}}]$, we note that:

$$\begin{aligned}
\mathbb{E}[\mathrm{x}_{\boldsymbol{w}}] &= \mathbb{E} \left[ \boldsymbol{\varepsilon}^\top \boldsymbol{\Phi} (\boldsymbol{w} - \hat{\boldsymbol{w}}) \right] \\
&= \mathbb{E} \left[ \boldsymbol{\varepsilon}^\top \boldsymbol{\Phi} \hat{\boldsymbol{w}} \right] \\
&\leq \sqrt{\mathbb{E} \left[ \|\boldsymbol{\varepsilon}^\top \boldsymbol{\Phi}\|_2^2 \right]} \sqrt{\mathbb{E} \left[ \|\hat{\boldsymbol{w}}\|_2^2 \right]} \\
&\leq \sigma \sqrt{\sum_{j=1}^{d} \sum_{i=1}^{n} |\phi_j(x_i)|^2} R \\
&= \sigma \sqrt{n} M R
\end{aligned} \tag{26}$$

Here, the third inequality follows from the Cauchy-Schwarz inequality, and the fourth inequality is derived from the fact that $|\boldsymbol{\varepsilon}^\top \boldsymbol{\Phi}_j|^2/(\sigma\|\boldsymbol{\Phi}_j\|_2)^2$ follows a chi-squared distribution with 1 degrees of freedom and $\hat{\boldsymbol{w}} \in \mathcal{V}_R$.

By combining Eq. (24), Eq. (25), and Eq. (26), we obtain the following bound with some constant $C$:

$$\Pr\left(\frac{2}{n}\sup_{\boldsymbol{w}\in\mathcal{V}_R}\mathrm{x}_{\boldsymbol{w}} \leq C\sigma M\Gamma R\left(\sqrt{\frac{d_\mathcal{V}\log d_\mathcal{V} d}{n}} + \sqrt{\frac{\log 2\beta}{n}} + 2\sqrt{\frac{\log(\delta/2)}{n}}\right)\right) \geq 1 - \delta.$$

This completes the proof. □

**Lemma D.1.** *Let $\hat{\boldsymbol{w}}$ be a minimizer of the following optimization problem:*

$$\hat{\boldsymbol{w}} = \arg\min_{\boldsymbol{w}\in\mathcal{V}_R} \frac{1}{n}\|\boldsymbol{y} - \boldsymbol{\Phi}\boldsymbol{w}\|_2^2, \tag{27}$$

*where $\mathcal{V}_R = \mathcal{V}(\mathscr{D}, \mathcal{B}, \mathcal{T}) \cap \mathbb{B}_2(R)$ is the affine variety constrained with the $\ell_2$-ball, $\boldsymbol{y} = \boldsymbol{\Phi}\boldsymbol{w}^* + \boldsymbol{\varepsilon}$ is the observed vector, $\boldsymbol{\Phi}$ is the design matrix, $\boldsymbol{w}^* \in \mathcal{V}_R$ is the true parameter vector, and $\boldsymbol{\varepsilon} = [\varepsilon_1, \ldots, \varepsilon_n]^\top$ is the noise vector with each $\varepsilon_i$ independently following a zero-mean Gaussian distribution. Then, under these conditions, we have:*

$$\|\boldsymbol{\Phi}(\boldsymbol{w}^* - \hat{\boldsymbol{w}})\|_2^2 \leq 2\boldsymbol{\varepsilon}^\top\boldsymbol{\Phi}(\boldsymbol{w}^* - \hat{\boldsymbol{w}}). \tag{28}$$

*Proof.* Since $\hat{\boldsymbol{w}}$ is a minimizer of Eq. (27), we have:

$$\|\boldsymbol{y} - \boldsymbol{\Phi}\hat{\boldsymbol{w}}\|_2^2 \leq \|\boldsymbol{y} - \boldsymbol{\Phi}\boldsymbol{w}^*\|_2^2 = \|\boldsymbol{\varepsilon}\|_2^2.$$

The left-hand side can be expanded as:

$$\|\boldsymbol{y} - \boldsymbol{\Phi}\hat{\boldsymbol{w}}\|_2^2 = \|\boldsymbol{y} - \boldsymbol{\Phi}\boldsymbol{w}^* + \boldsymbol{\Phi}\boldsymbol{w}^* - \boldsymbol{\Phi}\hat{\boldsymbol{w}}\|_2^2$$
$$= \|\boldsymbol{\varepsilon} - \boldsymbol{\Phi}(\boldsymbol{w}^* - \hat{\boldsymbol{w}})\|_2^2.$$

Thus, we have:

$$\|\boldsymbol{\varepsilon} - \boldsymbol{\Phi}(\boldsymbol{w}^* - \hat{\boldsymbol{w}})\|_2^2 \leq \|\boldsymbol{\varepsilon}\|_2^2.$$

Expanding the left-hand side, we get:

$$\|\boldsymbol{\varepsilon} - \boldsymbol{\Phi}(\boldsymbol{w}^* - \hat{\boldsymbol{w}})\|_2^2 = \|\boldsymbol{\varepsilon}\|_2^2 - 2\boldsymbol{\varepsilon}^\top\boldsymbol{\Phi}(\boldsymbol{w}^* - \hat{\boldsymbol{w}}) + \|\boldsymbol{\Phi}(\boldsymbol{w}^* - \hat{\boldsymbol{w}})\|_2^2.$$

Subtracting $\|\boldsymbol{\varepsilon}\|_2^2$ from both sides, we obtain:

$$\|\boldsymbol{\Phi}(\boldsymbol{w}^* - \hat{\boldsymbol{w}})\|_2^2 \leq 2\boldsymbol{\varepsilon}^\top\boldsymbol{\Phi}(\boldsymbol{w}^* - \hat{\boldsymbol{w}}).$$

This completes the proof. □

**Lemma D.2.** *Let $\mathrm{z}_{\boldsymbol{w}}$ be the zero-mean random process in the metric space $(\mathcal{V}_R, \|\cdot\|_2)$, which have the following sub-Gaussian increments. For all $\boldsymbol{w}_1, \boldsymbol{w}_2 \in \mathcal{V}_R$,*

$$\|\mathrm{z}_{\boldsymbol{w}_1} - \mathrm{z}_{\boldsymbol{w}_2}\|_{\psi_2} \leq A\|\boldsymbol{w}_1 - \boldsymbol{w}_2\|_2,$$

*where $\|\cdot\|_{\psi_2}$ is the sub-Gaussian norm, $A$ is a positive constant. Then, the expectation of the supremum of the process can be bounded as follows.*

$$\mathbb{E}\sup_{\boldsymbol{w}\in\mathcal{V}_R}\mathrm{z}_{\boldsymbol{w}} \leq CAR\left(\sqrt{d_\mathcal{V}\log d_\mathcal{V} d} + \sqrt{\log 2\beta}\right),$$

*where $C$ is positive constant.*

*Proof.* Using Dudley's integral inequality [17] to the zero-mean random process:

$$\mathbb{E}\sup_{\boldsymbol{w}\in\mathcal{V}_R}\mathrm{z}_{\boldsymbol{w}} \leq C_0 A \int_0^\infty \sqrt{\log\mathcal{N}(\mathcal{V}_R, \varepsilon, \|\cdot\|_2)}\mathrm{d}\varepsilon. \tag{29}$$

Since the set $\mathcal{V}_R$ is $(\beta, d_\mathcal{V})$ regular set from Lemma 2.13 by Zhang and Kileel [42], Lemma B.2 shows the upper bound of the covering number for any $\varepsilon \in (0, 2R]$ as follows.

$$\log\mathcal{N}(\mathcal{V}_R, \varepsilon, \|\cdot\|_2) \leq d_\mathcal{V}\log\left(\frac{2Rd_\mathcal{V} d}{\varepsilon}\right) + \log 2\beta.$$

We substitute the above inequality to Eq. (29):

$$\mathbb{E} \sup_{\boldsymbol{w} \in \mathcal{V}_R} z_{\boldsymbol{w}} \le C_0 A \left( \sqrt{d_{\mathcal{V}}} \int_0^\infty \sqrt{\log \left( \frac{2R d_{\mathcal{V}} d}{\varepsilon} \right)} \mathrm{d}\varepsilon + 2R\sqrt{\log 2\beta} \right).$$

The integral in the first term can be calculated using substitution and integration by parts. Let

$$I := \int_0^\infty \sqrt{\log \left( \frac{2R d_{\mathcal{V}} d}{\varepsilon} \right)} \mathrm{d}\varepsilon = \int_0^{2R} \sqrt{\log \left( \frac{2R d_{\mathcal{V}} d}{\varepsilon} \right)} \mathrm{d}\varepsilon.$$

We substitute $\chi := 2R d_{\mathcal{V}} d$, $u := \log(\chi/\varepsilon)$ into the integral:

$$I = \int_\infty^{\log d_{\mathcal{V}} d} u^{1/2} (-\chi e^{-u}) \mathrm{d}u.$$

To solve the above integral, we use the formula for integration by parts:

$$I = -\chi \left( [-u^{1/2} e^{-u}]_\infty^{\log d_{\mathcal{V}} d} + \frac{1}{2} \int_\infty^{\log d_{\mathcal{V}} d} u^{-1/2} e^{-u} \mathrm{d}u \right)$$

$$= 2R\sqrt{\log d_{\mathcal{V}} d} + R d_{\mathcal{V}} d \int_{\log d_{\mathcal{V}} d}^\infty u^{-1/2} e^{-u} \mathrm{d}u.$$

The integral in the second term can be upper bounded as follows.

$$\int_{\log d_{\mathcal{V}} d}^\infty u^{-1/2} e^{-u} \mathrm{d}u \le \int_{\log d_{\mathcal{V}} d}^\infty e^{-u} \mathrm{d}u = [-e^{-u}]_{\log d_{\mathcal{V}} d}^\infty = (d_{\mathcal{V}} d)^{-1}.$$

We obtain the following bound with some constant $C$.

$$\mathbb{E} \sup_{\boldsymbol{w} \in \mathcal{V}_R} z_{\boldsymbol{w}} \le CAR \left( \sqrt{d_{\mathcal{V}} \log d_{\mathcal{V}} d} + \sqrt{\log 2\beta} \right).$$

$\square$

## E  Proof for Proposition 3.7 and Proposition 4.4

**Proposition 3.7.** *The effective dimension of the PI kernel associated with the affine variety $\mathcal{V}(\mathscr{D}, \mathcal{B}, \mathcal{T}) = \{\boldsymbol{w} : \boldsymbol{D}\boldsymbol{w} = \boldsymbol{0}\}$ with dimension $d_{\mathcal{V}}$ is upper bounded by*

$$d_{\mathrm{eff}}(\xi, \nu) \lesssim \sum_{j=1}^{d_{\mathcal{V}}} \frac{1}{1+\xi} + \sum_{j=d_{\mathcal{V}}}^d \frac{1}{1+\xi+\nu\alpha_j} \le \frac{d}{1+\xi}.$$

*where $\{\alpha_j\}_{j=d_{\mathcal{V}}}^d$ denote the positive eigenvalues of the matrix $\boldsymbol{D}^\top \boldsymbol{T} \boldsymbol{D}$.*

*Proof.* From Theorem 4.2 in [15] and Equation 15 in [16], the effective dimension is bounded as follows:

$$d_{\mathrm{eff}}(\xi, \nu) \lesssim \sum_{\alpha \in \sigma(\boldsymbol{B}\boldsymbol{M}^{-1}\boldsymbol{B})} \frac{1}{1+\alpha^{-1}} \le \sum_{\alpha \in \sigma(\boldsymbol{M}^{-1})} \frac{1}{1+\alpha^{-1}}, \tag{30}$$

where $\boldsymbol{M} := \xi\boldsymbol{I} + \nu\boldsymbol{D}^\top\boldsymbol{T}\boldsymbol{D} \in \mathbb{R}^{d\times d}$ and $\boldsymbol{B} \in \mathbb{R}^{d\times d}$ is the Gram matrix of the basis functions, i.e., $B_{j,j'} = \langle \phi_j, \phi_{j'} \rangle_\mu$ for all $\phi_j, \phi_{j'} \in \mathcal{B}$.

Since the matrix $\boldsymbol{D}^\top\boldsymbol{T}\boldsymbol{D}$ is positive semi-definite, the eigenvalues of the matrix $\boldsymbol{M}$ in ascending order $\sigma_j(\cdot)$ are given by

$$\sigma_j(\boldsymbol{M}) = \left\{ \begin{array}{ll} \xi & (j = 1, \ldots, d_{\mathcal{V}}) \\ \xi + \nu\alpha_j & (d_{\mathcal{V}} < j) \end{array} \right. .$$

Therefore, the matrix $\boldsymbol{M}$ is positive definite, and the eigenvalues of $\boldsymbol{M}^{-1}$ are $\alpha^{-1}$ for all $\alpha \in \sigma(\boldsymbol{M})$. Combining this with Eq. (30), we obtain the first inequality. The second inequality is obtained when $\nu = 0$.

$\square$

**Proposition 4.4.** *Suppose the operator $\mathscr{D}$ can be decomposed as $\mathscr{D} = \mathscr{L} + \mathscr{F}$, where $\mathscr{L}$ is a nonzero linear differential operator and $\mathscr{F}$ is a nonlinear operator. Then, we have $d_{\mathcal{V}(\mathscr{L})} \leq d_{\mathcal{V}(\mathscr{D})}$.*

*Proof.* The point $\boldsymbol{w} = \boldsymbol{0}$ lies on $\mathcal{V}(\mathscr{D})$, and if $\mathscr{L} \neq 0$, it is not singular. The Jacobian rank of polynomials $p_k(\boldsymbol{w}) = \langle \mathscr{D}[\boldsymbol{w}^\top \boldsymbol{\phi}], \psi_k \rangle_{\mu_k}$ in $\boldsymbol{w} = \boldsymbol{0}$ is equal to $d - d_{\mathcal{V}(\mathscr{L})}$. By Definition 4.3, we have $d_{\mathcal{V}(\mathscr{L})} \leq d_{\mathcal{V}(\mathscr{D})}$. $\qquad\square$

# F  Minimax Risk Analysis for Physics-Informed Models with General Architectures via Rademacher Complexity

In this section, we extend our analysis to general model architectures parameterized by polynomial functions of the weights. Our primary objective is to establish a minimax risk framework grounded in Rademacher complexity. This allows us to handle richer hypothesis spaces while incorporating structural constraints imposed by physical laws.

## F.1  Notation and Definitions

To set the stage, we introduce several fundamental notions that will be used throughout the complexity analysis. We begin with norms for vector- and matrix-valued objects, which help measure the size and regularity of functions and parameters. These norms provide the foundation for bounding Rademacher complexity.

**Definition F.1** (Mixed Norm for Vector-Valued Functions). Let $f : \mathcal{X} \to \mathbb{R}^{d_{\text{out}}}$ be a vector-valued function. Its $(\infty, p)$-norm is defined by

$$\|f\|_{\infty,p} := \sup_{x \in \mathcal{X}} \|f(x)\|_p, \quad \text{where} \quad \|f(x)\|_p := \left( \sum_{i=1}^{d_{\text{out}}} |f_i(x)|^p \right)^{1/p}. \tag{31}$$

The above norm enables us to uniformly control the $p$-norm magnitude of the function outputs across the entire input domain.

**Definition F.2** (Matrix $p$-Norm). For a matrix $W \in \mathbb{R}^{d_{\text{in}} \times d_{\text{out}}}$, we regard $W$ as a vector $\text{vec}(W) \in \mathbb{R}^{d_{\text{in}} d_{\text{out}}}$. Its norm is defined using the standard vector $p$-norm:

$$\|W\|_p := \left( \sum_{i=1}^{m} \sum_{j=1}^{n} |W_{ij}|^p \right)^{1/p}, \quad (1 \leq p < \infty).$$

This definition allows us to consistently measure parameter magnitudes, regardless of whether they appear as vectors or matrices.

**Definition F.3** (Rademacher Complexity). Let $\mathcal{F}$ be a function class and $S = \{x_1, \ldots, x_n\}$ an i.i.d. sample. The empirical Rademacher complexity is defined as

$$\widehat{\mathfrak{R}}_S(\mathcal{F}) = \frac{1}{n} \mathbb{E}_{\boldsymbol{\tau}} \left[ \sup_{f \in \mathcal{F}} \left| \sum_{i=1}^{n} \tau_i \, f(x_i) \right| \right],$$

where $\tau_1, \ldots, \tau_n$ are independent Rademacher variables taking values in $\{\pm 1\}$. The Rademacher complexity is obtained by further taking the expectation of $\widehat{\mathfrak{R}}_S(\mathcal{F})$ over the random sample $S$.

Rademacher complexity serves as a central tool for quantifying the richness of hypothesis spaces and will be essential in deriving minimax risk bounds.

## F.2  Definition of the Hypothesis Space

We now define the hypothesis space of interest. To ensure well-posedness of our analysis, the class is required to satisfy boundedness and Lipschitz continuity conditions.

**Definition F.4** (Lipschitz Polynomial Hypothesis Space). Let $\mathcal{H}$ denote a hypothesis space consisting of functions $f_{\boldsymbol{w}} : \mathbb{R}^m \to \mathbb{R}$, parameterized by $\boldsymbol{w} \in \mathbb{R}^d$, where each $f_{\boldsymbol{w}}$ is polynomial in $\boldsymbol{w}$.

The parameter domain is restricted to

$$\boldsymbol{w} \in \mathcal{V}_R := \mathcal{V}(\mathscr{D}, \mathcal{T}) \cap \mathbb{B}_2(R), \tag{32}$$

where the affine variety $\mathcal{V}(\mathscr{D}, \mathcal{T})$ is given by

$$\mathcal{V}(\mathscr{D}, \mathcal{T}) := \left\{ \boldsymbol{w} \in \mathbb{R}^d : \langle \mathscr{D}[f_{\boldsymbol{w}}] - v, \psi_k \rangle_{\mu_k} = 0, \ \forall (\psi_k, \mu_k) \in \mathcal{T} \right\}. \tag{33}$$

Here, $\mathbb{B}_2(R) := \{\boldsymbol{w} \in \mathbb{R}^d : \|\boldsymbol{w}\|_2 \leq R\}$ denotes the Euclidean ball of radius $R$.

Furthermore, there exists a constant $\ell_{\mathcal{H}} > 0$ such that

$$\|f_{\boldsymbol{w}} - f_{\boldsymbol{w}'}\| \leq \ell_{\mathcal{H}} \|\boldsymbol{w} - \boldsymbol{w}'\|_2, \quad \forall \boldsymbol{w}, \boldsymbol{w}' \in \mathcal{V}_R, \tag{34}$$

ensuring Lipschitz continuity of the parameter-to-function mapping.

The above construction provides a general framework. Next, we highlight an important special case relevant to physics-informed neural networks (PINNs).

**Special Case: Polynomial PINN**

**Definition F.5** (Polynomial PINN Hypothesis Space). Let $\mathcal{H}_L$ denote the hypothesis space represented by a fully-connected neural network of depth $L$ with polynomial activation $\phi$:

$$f_{\boldsymbol{w}}(x) = W_L \phi(W_{L-1} \cdots \phi(W_1 x) \cdots),$$

with parameter vector $\boldsymbol{w} = \text{vec}(W_1, \ldots, W_L) \in \mathbb{R}^d$. The parameter domain is restricted to

$$\boldsymbol{w} \in \mathcal{V}_R = \mathcal{V}(\mathscr{D}, \mathcal{T}) \cap \mathbb{B}_2(R),$$

where $\mathcal{V}(\mathscr{D}, \mathcal{T})$ is the affine variety in equation 33.

To ensure the polynomial PINN setting remains mathematically well-posed, we introduce additional assumptions on boundedness and Lipschitz continuity.

**Assumption F.6** (Uniformly Bounded Target and Hypothesis Class). The true regression function $f^* : \mathbb{R}^m \to \mathbb{R}$ is uniformly bounded as $\|f^*\|_\infty \leq F_{\max}$. Moreover, every hypothesis $f_{\boldsymbol{w}} \in \mathcal{H}$ satisfies the same bound: $\|f_{\boldsymbol{w}}\|_\infty \leq F_{\max}$.

**Assumption F.7** (Lipschitz Continuity and Boundedness of Polynomial Activation). The polynomial activation function $\phi$ is uniformly bounded, $\|\phi\|_{\infty,2} \leq M_\phi$ for some constant $M_\phi > 0$. Moreover, $\phi$ is Lipschitz continuous with constant $L_\phi$, i.e., for any $z_1, z_2 \in \mathbb{R}$:

$$\|\phi(z_1) - \phi(z_2)\|_2 \leq L_\phi \|z_1 - z_2\|_2.$$

Finally, we show that under the above assumptions, polynomial PINNs inherit a Lipschitz property at the function level.

**Lemma F.8** (Lipschitz Property of Polynomial PINNs). *Suppose Assumption F.7 holds. For two polynomial PINNs $f_{\boldsymbol{w}}, f_{\boldsymbol{w}'} \in \mathcal{H}_L$ with parameters $\boldsymbol{w}, \boldsymbol{w}' \in \mathcal{V}_R$, the mapping from parameters to functions is Lipschitz continuous, i.e.,*

$$\|f_{\boldsymbol{w}} - f_{\boldsymbol{w}'}\|_\infty \leq \ell_{\mathcal{H}_L} \|\boldsymbol{w} - \boldsymbol{w}'\|_2, \tag{35}$$

*where the Lipschitz constant $\ell_{\mathcal{H}_L}$ depends on the network architecture as*

$$\ell_{\mathcal{H}_L} = M \frac{(RL_\phi)^{L-1} - 1}{RL_\phi - 1} + R(RL_\phi)^{L-1}. \tag{36}$$

*Consequently, the polynomial PINN hypothesis space $\mathcal{H}_L$ satisfies the Lipschitz condition in Definition F.4, and therefore*

$$\mathcal{H}_L \subseteq \mathcal{H}.$$

*Proof.* Let $h_\ell$ and $h'_\ell$ be the outputs of layer $\ell$ for parameters $\boldsymbol{w}$ and $\boldsymbol{w}'$ respectively. We can establish a recursive inequality:

$$\begin{aligned}
\|h_\ell - h'_\ell\|_{\infty,2} &= \|W_\ell \phi(h_{\ell-1}) - W'_\ell \phi(h'_{\ell-1})\|_{\infty,2} \\
&\leq \|W_\ell(\phi(h_{\ell-1}) - \phi(h'_{\ell-1}))\|_{\infty,2} + \|(W_\ell - W'_\ell)\phi(h'_{\ell-1})\|_{\infty,2} \\
&\leq \|W_\ell\|_2 L_\phi \|h_{\ell-1} - h'_{\ell-1}\|_{\infty,2} + \|W_\ell - W'_\ell\|_2 \|\phi(h'_{\ell-1})\|_{\infty,2} \\
&\leq RL_\phi \|h_{\ell-1} - h'_{\ell-1}\|_{\infty,2} + \varepsilon M.
\end{aligned}$$

Solving this recurrence relation for $\|h_L - h'_L\|_\infty = \|f_{\boldsymbol{w}} - f_{\boldsymbol{w}'}\|_\infty$ yields the constant $\ell_L$. The relationship between covering numbers follows directly. $\qquad\square$

### F.3 Generalization Bound

Based on the assumptions, we can now control the Rademacher complexity of the Lipschitz polynomial hypothesis class $\mathcal{H}$ through a Dudley integral bound.

**Lemma F.9** (Dudley Integral Bound for Physics-Informed Models). *Let $\mathcal{H}$ be the hypothesis space defined in Definition F.4, where the underlying affine variety $\mathcal{V}$ is $(\beta, d_{\mathcal{V}})$-regular. Then the Rademacher complexity of $\mathcal{H}$ is bounded as*

$$\mathfrak{R}(\mathcal{H}_L) \leq C\, F_{\max}\, \ell_{\mathcal{H}} \left( \sqrt{\frac{d_{\mathcal{V}}}{n}\, \ln\left(\frac{2R\,d_{\mathcal{V}}\,d}{\ell_{\mathcal{H}}\,F_{\max}}\right)} + \sqrt{\frac{\ln(2\beta)}{n}} \right), \tag{37}$$

*for some constant $C > 0$.*

*Proof.* First, for any $\boldsymbol{w}_1, \boldsymbol{w}_2 \in \mathcal{V}_R$ and corresponding $f_{\boldsymbol{w}_1}, f_{\boldsymbol{w}_2} \in \mathcal{H}$, Lipsitz continuity reads
$$\|\boldsymbol{w}_1 - \boldsymbol{w}_2\| \leq \varepsilon \implies \|f_{\boldsymbol{w}_1} - f_{\boldsymbol{w}_2}\| \leq \varepsilon \ell_{\mathcal{H}}$$

Hence
$$\mathcal{N}\left(\mathcal{H}, \varepsilon M, \|\cdot\|_1\right) \leq \mathcal{N}\left(\mathcal{V}_R, \varepsilon, \|\cdot\|\right).$$

By Dudley's integral bound and Lemma B.2 on $\ell_p$–covers,

$$\begin{aligned}
\mathfrak{R}(\mathcal{H}) &\leq \frac{12}{\sqrt{n}} \int_0^{F_{\max}} \sqrt{\ln \mathcal{N}\left(\mathcal{H}, \varepsilon, \|\cdot\|_1\right)}\, d\varepsilon \\
&\leq \frac{12}{\sqrt{n}} \int_0^{F_{\max}\ell_{\mathcal{H}}} \sqrt{\ln \mathcal{N}\left(\mathcal{V}_{p,R}, \varepsilon, \|\cdot\|_p\right)}\, d\varepsilon \\
&= 12 \int_0^{F_{\max}\ell_{\mathcal{H}}} \sqrt{\frac{d_{\mathcal{V}}}{n}\, \ln\left(\frac{2R\,d_{\mathcal{V}}\,d}{\varepsilon}\right)}\, d\varepsilon + F_{\max}\ell_{\mathcal{H}}\sqrt{\frac{\ln(2\beta)}{n}}.
\end{aligned}$$

Define

$$I = \int_0^{F_{\max}\ell_{\mathcal{H}}} \sqrt{\ln\left(\frac{2R\,d_{\mathcal{V}}\,d}{\varepsilon}\right)}\, d\varepsilon.$$

Similar calculation in Lemma D.2 shows

$$I \leq F_{\max}\ell_{\mathcal{H}}\sqrt{\ln\left(\frac{2R\,d_{\mathcal{V}}\,d}{\ell_{\mathcal{H}}\,F_{\max}}\right)}.$$

Combining these estimates yields the stated bound. $\qquad\square$

**Lemma F.10** (Maximum of sub-exponential random variables). *Let $X_1, \ldots, X_n$ be independent, identically distributed sub-exponential random variables satisfying $\|X_i\|_{\psi_1} \leq \nu$ for every $i \in \{1, \ldots, n\}$. Then there exists an absolute constant $C > 0$ such that*

$$\left\| \max_{1 \leq i \leq n} |X_i| \right\|_{\psi_1} \leq C\,\nu \log n.$$

*Proof.* From the assumption $\|X_i\|_{\psi_1} \leq \nu$, we have the standard sub-exponential tail bound:

$$\Pr\left\{ \max_{1 \leq i \leq n} X_i \geq t \right\} \leq 2\exp\left( -\frac{1}{2}\min\left\{ \frac{t^2}{\nu^2}, \frac{t}{\nu} \right\} \right).$$

Moreover, for all $t \geq 0$, we can uniformly bound the maximum via a union bound:

$$\begin{aligned}
\Pr\left\{ \max_{1 \leq i \leq n} X_i \geq t \right\} = \Pr\left[ \bigcup_{i=1}^n \{X_i \geq t\} \right] &\leq \sum_{i=1}^n \Pr\{X_i \geq t\} \\
&\leq 2n\exp\left( -\min\left\{ \frac{t^2}{2\nu^2}, \frac{t}{2\nu} \right\} \right). \tag{38}
\end{aligned}$$

Now consider two cases based on the relation between $n$ and the exponent.

**Case 1:** Suppose

$$n \leq \exp\left(\frac{1}{2}\min\left\{\frac{t^2}{2\nu^2}, \frac{t}{2\nu}\right\}\right).$$

Then, by inequality equation 38, we have

$$\Pr\left\{\max_i X_i \geq t\right\} \leq 2\exp\left(-\frac{1}{2}\min\left\{\frac{t^2}{2\nu^2}, \frac{t}{2\nu}\right\}\right)$$

$$\leq 2\exp\left(-\frac{1}{3\log n}\min\left\{\frac{t^2}{2\nu^2}, \frac{t}{2\nu}\right\}\right),$$

which already provides a stronger tail decay than we ultimately require.

**Case 2:** Suppose

$$n > \exp\left(\frac{1}{2}\min\left\{\frac{t^2}{2\nu^2}, \frac{t}{2\nu}\right\}\right).$$

Then, the probability on the right-hand side of Equation (38) satisfies

$$\Pr\left\{\max_{1\leq i\leq n} X_i \geq t\right\} \leq 2\exp\left(-\frac{1}{3\log n}\min\left\{\frac{t^2}{2\nu^2}, \frac{t}{2\nu}\right\}\right) > 2e^{-2/3} > 1,$$

which is vacuously bounded above by 1. Thus, it does not affect the validity of our bound.

In either case, we obtain the uniform upper bound

$$\Pr\left\{\max_{1\leq i\leq n} X_i \geq t\right\} \leq 2\exp\left(-\frac{1}{3\log n}\min\left\{\frac{t^2}{2\nu^2}, \frac{t}{2\nu}\right\}\right)$$

$$= 2\exp\left(-\frac{1}{3}\min\left\{\frac{t^2}{2(\nu\sqrt{\log n})^2}, \frac{t}{2(\nu\log n)}\right\}\right).$$

This tail bound implies that

$$\left\|\max_{1\leq i\leq n} X_i\right\|_{\psi_1} \leq C\nu\log n,$$

for some universal constant $C > 0$. $\qquad\square$

**Lemma F.11.** *Let* $g_{\boldsymbol{w}}(x,y) := (f_{\boldsymbol{w}}(x) - y)^2 - \mathbb{E}_{(X,Y)}\left[(f_{\boldsymbol{w}}(X) - Y)^2\right]$ *for* $f_{\boldsymbol{w}} \in \mathcal{H}$. *Under Assumption F.6 assume the noise* $\epsilon = Y - f^*(X)$ *is sub-Gaussian with proxy variance* $\sigma^2$. *Then there exists a constant* $C > 0$ *(independent of* $\boldsymbol{w}$*) such that*

$$\|g_{\boldsymbol{w}}\|_{\psi_1} \leq 2(4F_{\max} + \sigma)^2, \qquad \sup_{\boldsymbol{w}} \mathbb{E}\left[g_{\boldsymbol{w}}(X,Y)^2\right] \leq C(4F_{\max} + \sigma)^4$$

*Proof.* By the triangle inequality for the sub-Gaussian norm,

$$\|f_{\boldsymbol{w}}(X) - Y\|_{\psi_2} \leq \|f_{\boldsymbol{w}}(X) - f^*(X)\|_{\psi_2} + \|\epsilon\|_{\psi_2}$$

$$\leq 4F_{\max} + \sigma$$

Using the identity $\|Z^2\|_{\psi_1} = \|Z\|_{\psi_2}^2$ for any $Z$, we get

$$\|(f_{\boldsymbol{w}}(X) - Y)^2\|_{\psi_1} \leq (4F_{\max} + \sigma)^2.$$

By the triangle inequality in the $\psi_1$–norm,

$$\|g_{\boldsymbol{w}}(X,Y)\|_{\psi_1} \leq \|(f_{\boldsymbol{w}}(X) - Y)^2\|_{\psi_1} + \|\mathbb{E}(f_{\boldsymbol{w}}(X) - Y)^2\|_{\psi_1}$$

$$\leq 2(4F_{\max} + \sigma)^2.$$

Finally, to bound the second moment,

$$\mathbb{E}\left[g_{\boldsymbol{w}}(X,Y)^2\right] = \mathrm{Var}\left(g_{\boldsymbol{w}}(X,Y)\right) \leq C(4F_{\max} + \sigma)^4$$

for suitable constant $C > 0$. This completes the proof. $\qquad\square$

**Lemma F.12.** *Let $\ell(u, y) = (u - y)^2$. The Rademacher complexity of the composite class $\ell \circ \mathcal{H} = \{(f_{\boldsymbol{w}}(x) - y)^2 : f_{\boldsymbol{w}} \in \mathcal{H}\}$ satisfies*

$$\mathfrak{R}(\ell \circ \mathcal{H}) \leq \left(4F_{\max} + 2\sigma\sqrt{2/\pi}\right)\mathfrak{R}(\mathcal{H}),$$

*where $F_{\max}$ is a uniform bound on $|f_{\boldsymbol{w}}(x)|$ and $\sigma^2$ is the noise variance.*

*Proof.* Define for each example $(x_i, y_i)$ the function

$$\phi_i(u) = (u - y_i)^2.$$

For any $u, v \in \mathbb{R}$,

$$\begin{aligned}
|\phi_i(u) - \phi_i(v)| &= \left|u^2 - v^2 + 2y_i\,(v - u)\right| \\
&= |(u - v)\,(u + v - 2y_i)| \\
&\leq (|u| + |v| + 2|y_i|)\,|u - v|\,.
\end{aligned}$$

Since $|f_{\boldsymbol{w}}(x)| \leq F_{\max}$ for all $x$ and $|y_i| \leq F_{\max} + |\varepsilon_i|$, it follows that

$$|\phi_i(u) - \phi_i(v)| \leq (4F_{\max} + 2|\varepsilon_i|)\,|u - v|\,.$$

Applying the Rademacher contraction lemma to the empirical Rademacher complexity $\widehat{\mathfrak{R}}_S$,

$$\begin{aligned}
\mathfrak{R}(\ell \circ \mathcal{H}) &= \mathbb{E}_{X, \varepsilon_{1:n}}\left[\widehat{\mathfrak{R}}_S(\ell \circ \mathcal{H})\right] \\
&\leq \mathbb{E}_{X, \varepsilon}\left[(4F_{\max} + 2|\varepsilon|)\,\widehat{\mathfrak{R}}_S(\mathcal{H})\right] \\
&= (4F_{\max} + 2\,\mathbb{E}[|\varepsilon|])\,\mathbb{E}_X\left[\widehat{\mathfrak{R}}_S(\mathcal{H})\right].
\end{aligned}$$

Since $\mathbb{E}_\varepsilon[|\varepsilon|] \leq \sigma\sqrt{2/\pi}$ for Gaussian noise,

$$\mathfrak{R}(\ell \circ \mathcal{H}) \leq \left(4F_{\max} + 2\sigma\sqrt{2/\pi}\right)\mathfrak{R}(\mathcal{H}).$$

This completes the proof. $\qquad\qquad\qquad\qquad\qquad\qquad\qquad\qquad\qquad\qquad\qquad\qquad\quad\square$

Finally, combining the Rademacher complexity bound with Adamczak's concentration inequality [2] yields the following generalization bound for general physics-informed architectures.

**Theorem F.13** (Generalization Bound for Physics-Informed Models). *Let $f_{\boldsymbol{w}} \in \mathcal{H}$ be the Lipschitz polynomial hypothesis defined by Definition F.4, and assume that Assumption F.6 holds. Then there exist constants $C_0, C_1, C_2 > 0$ such that, with probability at least $1 - \delta$,*

$$\sup_{\boldsymbol{w} \in \mathcal{V}_R} |\mathcal{L}_n(\boldsymbol{w}) - \mathcal{L}(\boldsymbol{w})| \tag{39}$$

$$\leq C_0\left(4F_{\max} + \sigma\sqrt{\frac{2}{\pi}}\right)F_{\max}\,\ell_{\mathcal{H}}\left(\sqrt{\frac{d_{\mathcal{V}}}{n}\,\ln\left(\frac{2R\,d_{\mathcal{V}}\,d}{\ell_{\mathcal{H}}\,F_{\max}}\right)} + \sqrt{\frac{\ln(2\beta)}{n}}\right)$$

$$+ \max\left\{\sqrt{\frac{C_1(4F_{\max} + \sigma)^4}{n}\log\frac{4}{\delta}},\ \frac{C_2(4F_{\max} + \sigma)^2\log n}{n}\log\frac{12}{\delta}\right\}. \tag{40}$$

*Proof.* Apply Adamczak's concentration inequality [2] to the supremum:

$$\begin{aligned}
\Pr\left(\sup_w\left|\frac{1}{n}\sum_{i=1}^n g_w(X_i, Y_i)\right| \leq C\mathbb{E}\sup_w\left|\frac{1}{n}\sum_{i=1}^n g_w(X_i, Y_i)\right| + t\right) & \\
\geq 1 - \left(\exp\left(-\frac{t^2 n}{\widetilde{C}_1 V}\right) + \exp\left(-\frac{tn}{\nu}\right)\right), &
\end{aligned} \tag{41}$$

where $C$ is constant, $V$ and $\nu$ are quantities defined by

$$V := \sup_w \mathbb{E}[g_w(X, Y)^2],$$

$$\nu := \|\max_{1 \leq i \leq n} \sup_w g_w(X_i, Y_i)\|_{\psi_1}.$$

From Lemmas F.10 and F.11, we have

$$\sup_w \mathbb{E}[g_w(X,Y)^2] \leq C_1/\tilde{C}_1(4F_{\max} + \sigma)^4,$$

$$\| \max_{1 \leq i \leq n} \sup_w g_w(X_i, Y_i) \|_{\psi_1} \leq C_2(4F_{\max} + \sigma)^2 \log n,$$

where $C_1, C_2$ are constants.

By substituting the above quantities into inequality Equation (41) and converting it into a high-probability bound, we obtain, with probability at least $1 - \delta$, the following result:

$$
\begin{aligned}
\sup_w & \left| \frac{1}{n} \sum_{i=1}^n g_w(X_i, Y_i) \right| \\
&\leq C\mathbb{E} \sup_w \left| \frac{1}{n} \sum_{i=1}^n g_w(X_i, Y_i) \right| \\
&\quad + \max \left\{ \sqrt{\frac{C_1(4F_{\max} + \sigma)^4}{n} \log \frac{4}{\delta}}, \; \frac{C_2(4F_{\max} + \sigma)^2 \log n}{n} \log \frac{12}{\delta} \right\}.
\end{aligned}
\tag{42}
$$

We bound the expectation via symmetrization and Rademacher complexity:

$$
\begin{aligned}
\mathbb{E} \sup_w \left| \frac{1}{n} \sum_{i=1}^n g_w(X_i, Y_i) \right| &= \mathbb{E} \sup_w \left| \frac{1}{n} \sum_{i=1}^n \left( (f(X_i) - Y_i)^2 - (f(X_i') - Y_i')^2 \right) \right| \\
&= \mathbb{E} \sup_w \left| \frac{1}{n} \sum_{i=1}^n \tau_i \left( (f(X_i) - Y_i)^2 - (f(X_i') - Y_i')^2 \right) \right| \\
&\leq 2\mathbb{E} \sup_w \left| \frac{1}{n} \sum_{i=1}^n \tau_i (f(X_i) - Y_i)^2 \right| \\
&= 2\mathfrak{R}(\ell \circ \mathcal{H}),
\end{aligned}
\tag{43}
$$

where $\tau_i$ are Rademacher variables and $\mathfrak{R}(\ell \circ \mathcal{H})$ denotes the Rademacher complexity of the squared loss class.

Combining Lemmas F.9 and F.12 with Equations (42) and (43), these bounds complete the proof. $\square$

*Remark* F.14 (Limitations of Theorem F.13). While Theorem F.13 provides a theoretical generalization bound for polynomial PINNs with neural network surrogates, two caveats are worth emphasizing. First, the affine variety induced by the hard constraints is defined by a system of high-degree polynomial equations. The resulting algebraic structure is computationally intractable to characterize explicitly, which limits both theoretical validation and practical implementation. Second, recent findings suggest that the generalization behavior of over-parameterized models such as neural networks is not governed solely by the geometry of the hypothesis space, but is strongly affected by the implicit bias of the optimization algorithm. Hence, bounds of the form in Theorem F.13 may substantially deviate from the empirically observed performance.

# G   Experimental Detail

## G.1   Experiments on Strong Solution

In the experiments in Section 5.1, strong solutions to the equations are obtained analytically. The analytical solution with added Gaussian noise was used as data, the variance of the Gaussian noise was set to 0.01. The hyperparameters $L^2$ regularization weights and differential equation constraint weights $\xi$ and $\nu$ were searched in the range [1e-9, 1e-2] using the Optuna library [4]. The configuration with the smallest MSE on the validation data among 100 candidates was selected. All experiments were conducted on a MacBook Air equipped with an Apple M3 chip and 64 GB of unified memory. No external GPU or cluster computing resources were used.

**Harmonic Oscillator:** The initial value problem of a harmonic oscillator $\mathscr{D}[y] = 0$ with spring constant $k_s$ and mass $m_s$ on the domain $\Omega = [0, T]$ is given by:

$$\mathscr{D}[y] = \frac{\mathrm{d}^2}{\mathrm{d}t^2}y + \frac{k_s}{m_s}y, \quad y(0) = y_0, \quad \frac{\mathrm{d}}{\mathrm{d}t}y(0) = v_0.$$

We set the parameters $m_s = k_s = 1.0$, $T = 2\pi$. The initial position and velocity $[y_0, v_0]^\top$ are generated from the normal distribution $\mathcal{N}(\mathbf{1}, I)$, where $\mathbf{1}$ is an all-ones vector and $I$ is the identity matrix. The solution to the initial value problem is analytically given by:

$$y(t) = y_0 \cos(\omega t) + \frac{v_0}{\omega} \sin(\omega t), \quad \omega = \sqrt{k_s/m_s}.$$

The settings for the basis functions and the trial functions with the measure $\phi_j \in \mathcal{B}$, $(\psi_k, \mu_k) \in \mathcal{T}$ are as follows:

$$\phi_1(x) = 1, \ \phi_{2j}(x) = \cos\left(\frac{2\pi j}{T}x\right), \ \phi_{2j+1}(x) = \sin\left(\frac{2\pi j}{T}x\right) \ (j = 1, \ldots, d_t),$$
$$\psi_k(x) = 1, \ \mu_k = \delta_{x_k} \ (k = 1, \ldots, K),$$

where $d_t \in \{2, 4, 8, 16\}$ is the set of the number of basis functions, and $x_k \in \Omega$ is uniformly sampled from data with $K = 100$.

**Diffusion Equation:** The initial value problem for the one-dimensional diffusion equation $\mathscr{D}[u] = 0$ with diffusion coefficient $c$ and periodic boundary conditions is given by:

$$\mathscr{D}[u] = \frac{\partial}{\partial t}u - c\frac{\partial^2}{\partial x^2}u \quad (x, t) \in [-\Xi, \Xi] \times [0, T], u(x, 0) = u_0(x) \qquad x \in [-\Xi, \Xi]$$
$$u(-\Xi, t) = u(\Xi, t), \quad \frac{\partial u}{\partial x}(-\Xi, t) = \frac{\partial u}{\partial x}(\Xi, t).$$

We set the parameters $c = 1.0$, $\Xi = \pi$, $T = 2\pi$. The initial value $u_0$ is given by:

$$u_0(x) = \sum_{j=0}^{j_{\max}} A_j \cos(\omega_j x) + B_j \sin(\omega_j x), \ \omega_j = \frac{j\pi}{\Xi}, \tag{44}$$

where $[A_j, B_j]^\top$ are generated from the normal distribution $\mathcal{N}(\mathbf{1}, I)$ for all $j = 0, \ldots, j_{\max}$ and $j_{\max}$ is set to 1. The solution to the initial value problem is analytically given by:

$$u(x, t) = \sum_{j=0}^{j_{\max}} [A_j \cos(\omega_j x) + B_j \sin(\omega_j x)] e^{-c\omega_j^2 t}.$$

The settings for the basis functions and the trial functions with the measure $\phi_j \in \mathcal{B}$, $(\psi_k, \mu_k) \in \mathcal{T}$ are as follows:

$$\phi_1(x, t) = 1, \ \phi_{2j, j'}(x, t) = \cos(\omega_j x)e^{-c\omega_{j'}^2 t}, \ \phi_{2j+1, j'}(x, t) = \sin(\omega_j x)e^{-c\omega_{j'}^2 t}$$
$$(j = 1, \ldots, d_x, \ j' = 1, \ldots, d_t),$$
$$\psi_k(x, t) = 1, \ \mu_k = \delta_{(x_k, t_k)} \ (k = 1, \ldots, K),$$

where $d_t = 2$, $d_x \in \{10, 15, 20, 25\}$ are the sets of the number of basis functions, and $(x_k, t_k) \in \Omega$ is uniformly sampled from data with $K = 50 \times 500$.

## G.2 Experiments on Numerical Solution

In the experiments in Section 5.3, we numerically simulate the Bernoulli equation using the explicit Euler method and the diffusion equation using the finite difference method (FDM). The data used are the numerical solutions with added Gaussian noise of variance 0.01. The method for hyperparameter search is the same as described in Appendix G.1. For the nonlinear equations, we use the Adam optimizer with a learning rate of $1 \times 10^{-2}$, along with an exponential learning rate scheduler. The training is performed for a maximum of 2000 epochs, utilizing an early stopping technique.

**Discrete Bernoulli Equation:** The discrete Bernoulli equation $\mathscr{D}_h[y] = 0$ with the step size $h$ on the domain $\Omega = [0, T]$ is given by:

$$\mathscr{D}_h[y] = \frac{y_{\tau+1} - y_\tau}{h} + Py_\tau - Qy_\tau^\rho,$$

where $y_\tau = y(t_\tau)$ and $y_{\tau+1} = y(t_\tau + h)$ are evaluations on the grid $\{t_\tau\}_{\tau=1}^{n_t}$ with $n_t = \frac{T}{h}$. We set the constant parameters $(P, Q, \rho)$ to $(1.0, 0.0, 0.0)$ for the linear case and to $(1.0, 0.5, 2.0)$ for the non-linear case. We use varying $n_t \in \{100, 200\}$ with $T = 1.0$ for both cases. The initial state $y_0$ is generated from the standard normal distribution $\mathcal{N}(0, 1)$ for both cases. The ground-truth solution to the initial value problem is numerically solved by the explicit Euler method with step size $h$. The settings for the basis functions and the trial functions with measure $\phi_\tau \in \mathcal{B}_h$, $(\psi_\tau, \mu_\tau) \in \mathcal{T}_h$ are as follows:

$$\phi_\tau(t) = \begin{cases} 1 & \text{if } t \in [t_\tau, t_{\tau+1}) \\ 0 & \text{otherwise} \end{cases} \quad (\tau = 1, \dots, n_t),$$

$$\psi_\tau(t) = \phi_\tau(t), \quad \mu_\tau = \delta_{t_\tau} \quad (\tau = 1, \dots, n_t),$$

where $n_t = \frac{T}{h}$ is the same as the number of basis and trial functions, corresponding to the ground-truth solutions.

**Discrete Diffusion Equation:** The one-dimensional discrete diffusion equation $\mathscr{D}_h[u] = 0$ with the step size $h = [h_t, h_x]^\top$ and the diffusion coefficient $c(u)$ on the domain $\Omega = [-\Xi, \Xi] \times [0, T]$ is given by:

$$\mathscr{D}_h[u] = \frac{u_j^{\tau+1} - u_j^\tau}{h_t} - c(u_j^\tau)\frac{u_{j+1}^\tau - 2u_j^\tau + u_{j-1}^\tau}{h_x^2},$$

where $u_j^\tau := u(x_j, t_\tau)$, $u_j^{\tau+1} := u(x_j, t_\tau + h_t)$, and $u_{j\pm1}^\tau := u(x_j \pm h_x, t_\tau)$ are evaluations on the $n_x \times n_t$ size grid $\{x_j\}_{j=1}^{n_x} \times \{t_\tau\}_{\tau=1}^{n_t}$, where $n_x := \frac{2\Xi}{h_x}$ and $n_t := \frac{T}{h_t}$. The periodic boundary condition is adopted in the spatial domain, i.e., $u_{n_x+j}^\tau = u_j^\tau$ for any $j \in [d]$. The diffusion coefficient $c(u) = 1.0$ is used for the linear case and $c(u) = 0.1/(1 + u^2)$ for the nonlinear case. We use varying $(n_t, n_x) \in \{(400, 10), (400, 20), (400, 30)\}$ with $\Xi = 1.0$ and $T = 1.0$ for both cases. The initial value is generated with the same setting as shown in Eq. (44). The ground-truth solution to the initial value problem is numerically solved by the FDM with step sizes $h_t$ for the time domain and $h_x$ for the spatial domain. The settings for the basis functions and the trial functions with measure $\phi_{j,\tau} \in \mathcal{B}_h$, $(\psi_{j,\tau}, \mu_{j,\tau}) \in \mathcal{T}_h$ are as follows:

$$\phi_{j,\tau}(x, t) = \begin{cases} 1 & \text{if } (x, t) \in [x_j, x_{j+1}] \times [t_\tau, t_{\tau+1}] \\ 0 & \text{otherwise} \end{cases} \quad (j = 1, \dots, n_x, \ \tau = 1, \dots, n_t),$$

$$\psi_{j,\tau}(x, t) = \phi_{j,\tau}(x, t), \quad \mu_{j,\tau} = \delta_{(x_j, t_\tau)} \quad (j = 1, \dots, n_x, \ \tau = 1, \dots, n_t),$$

where $n_x = \frac{2\Xi}{h_x}$ and $n_t = \frac{T}{h_t}$ are the same as the number of basis and trial functions, corresponding to the ground-truth solutions.

## H   Additional Experimental Results

For each benchmark (discrete linear/nonlinear Bernoulli and Heat equations), we fix the number of basis functions $d$ and vary the size of the trial-function set $\mathcal{T}$. Reducing $|\mathcal{T}|$ relaxes the algebraic constraints on the learned solution, which in turn increases the dimension of the associated affine variety $d_\mathcal{V}$. As reported in Tables Figs. 4 and 5, when $|\mathcal{T}|$ decreases (and thus $d_\mathcal{V}$ increases), the Test MSE steadily increases. This consistent rising trend of Test MSE with larger $d_\mathcal{V}$ demonstrates that models endowed with fewer trial functions (i.e. weaker constraints) generalize more poorly.

(a) Linear Bernoulli eq.

| Settings | $h$ | 1/100 | | |
|---|---|---|---|---|
| Dimensions | $d$ | 100 | | |
| | $d_{\mathcal{V}}$ | 10 | 20 | 40 |
| Test MSE (PILR) | | $0.012 \pm 0.0023$ | $0.13 \pm 0.082$ | $0.33 \pm 0.22$ |

(b) Nonlinear Bernoulli eq.

| Settings | $h$ | 1/100 | | |
|---|---|---|---|---|
| Dimensions | $d$ | 100 | | |
| | $d_{\mathcal{V}}$ | 10 | 20 | 40 |
| Test MSE (PILR) | | $0.17 \pm 0.11$ | $0.21 \pm 0.14$ | $0.33 \pm 0.23$ |

Figure 4: Experimental results for PILR on the discrete Bernoulli equations.

(a) Linear Heat eq.

| Settings | $(h_t, h_x)$ | (1/400, 2/10) | | |
|---|---|---|---|---|
| Dimensions | $d$ | 4010 | | |
| | $d_{\mathcal{V}}$ | 110 | 210 | 410 |
| Test MSE (PILR) | | $1.6 \pm 0.35$ | $1.9 \pm 0.44$ | $2.0 \pm 0.49$ |

(b) Nonlinear Heat eq.

| Settings | $(h_t, h_x)$ | (1/200, 2/10) | | |
|---|---|---|---|---|
| Dimensions | $d$ | 2010 | | |
| | $d_{\mathcal{V}}$ | 110 | 210 | 410 |
| Test MSE (PILR) | | $0.37 \pm 0.11$ | $0.43 \pm 0.14$ | $0.56 \pm 0.19$ |

Figure 5: Experimental results for PILR on the discrete Heat equations.

# NeurIPS Paper Checklist

The checklist is designed to encourage best practices for responsible machine learning research, addressing issues of reproducibility, transparency, research ethics, and societal impact. Do not remove the checklist: **The papers not including the checklist will be desk rejected.** The checklist should follow the references and follow the (optional) supplemental material. The checklist does NOT count towards the page limit.

Please read the checklist guidelines carefully for information on how to answer these questions. For each question in the checklist:

- You should answer [Yes] , [No] , or [NA] .
- [NA] means either that the question is Not Applicable for that particular paper or the relevant information is Not Available.
- Please provide a short (1–2 sentence) justification right after your answer (even for NA).

**The checklist answers are an integral part of your paper submission.** They are visible to the reviewers, area chairs, senior area chairs, and ethics reviewers. You will be asked to also include it (after eventual revisions) with the final version of your paper, and its final version will be published with the paper.

The reviewers of your paper will be asked to use the checklist as one of the factors in their evaluation. While "[Yes] " is generally preferable to "[No] ", it is perfectly acceptable to answer "[No] " provided a

proper justification is given (e.g., "error bars are not reported because it would be too computationally expensive" or "we were unable to find the license for the dataset we used"). In general, answering "[No] " or "[NA] " is not grounds for rejection. While the questions are phrased in a binary way, we acknowledge that the true answer is often more nuanced, so please just use your best judgment and write a justification to elaborate. All supporting evidence can appear either in the main paper or the supplemental material, provided in appendix. If you answer [Yes] to a question, in the justification please point to the section(s) where related material for the question can be found.

IMPORTANT, please:

- **Delete this instruction block, but keep the section heading "NeurIPS Paper Checklist",**
- **Keep the checklist subsection headings, questions/answers and guidelines below.**
- **Do not modify the questions and only use the provided macros for your answers.**

