# OpenReview forum: "Understanding Generalization in Physics Informed Models through Affine Variety Dimensions"
_NeurIPS.cc/2025/Conference — NeurIPS 2025 poster_

### Official Review · Reviewer_qf6r · 2025-06-30

**Clarity:** 2
**Significance:** 3
**Originality:** 4
**Rating:** 5
**Confidence:** 3

**Summary:**

This paper presents a novel theoretical framework for analyzing the generalization performance of physics-informed regression models based on the affine variety induced by the PDE constraints. The main theorem of the paper identifies the dimension of the affine variety as the dominant factor governing generalization in physics informed models, rather than the number of model parameters. The discrete weak framework adopted also holds for both collocation and variational methods, allowing hybrid learning settings which are common in the physics-informed field. The authors provide a rigorous minimax risk analysis, propose a tractable approximation method for computing variety dimension, and empirically validate the theory using synthetic ODE/PDE benchmarks.

**Questions:**

I am open to raise my score in case the authors address the following questions.

* Can the authors provide further intuition on how their study applies to more common architectures as neural networks? In principle the approach of solving a least squares problem to find the estimator is similar to the idea behind extreme learning machines (which also exists in the physics-informed version). Can the theory in your article be applied to such a case?
* What is the practical outcome of Theorem 3.5? Can the authors clarify, based on the aforementioned theorem, what should be good practices for building/training physics-informed models?
* For the case of PINNs, the training is typically performed on a dense set of collocation points, in order to cover properly the whole training domain. How relevant is ensuring good generalization in these scenarios?

**Ethical Concerns:**

["NO or VERY MINOR ethics concerns only"]

**Final Justification:**

The discussion with the authors shed a positive light on their already good submission. I recommend acceptance for the paper.

**Limitations:**

The authors addressed the limitations of the paper.

**Paper Formatting Concerns:**

I have no concerns on the paper formatting.

**Quality:**

3

**Strengths And Weaknesses:**

I consider the following as key strengths of the paper:
* The central insight of the study is quite interesting and can be used to explain several performance issues of physics-informed models on specific PDEs (as Navier-Stokes).
* The paper connects concepts from statistical learning theory and algebraic geometry in very intuitive and elegant way.
* The paper is quite clear despite the intrinsic complexity of the theoretical framework

However, I feel the paper has two major flaws:
* The majority of the discussions in the paper are focusing on the PILR which is linear. This hinder the applicability of the study to architecture as PINNs, or other more common architectures in the physics-informed machine learning field.
* While the theoretical outcome is interesting per-se, I feel that the paper lacks some practical insights based on the theory.

---

> ### Author Rebuttal · Authors · 2025-07-29
>
> We greatly appreciate the time and expertise you invested in reviewing our work. Your comments provided important guidance for our revisions.
>
> > Q1: Can the authors provide further intuition on how their study applies to more common architectures as neural networks? In principle the approach of solving a least squares problem to find the estimator is similar to the idea behind extreme learning machines (which also exists in the physics-informed version). Can the theory in your article be applied to such a case?
>
> We thank the reviewer for this insightful question.
>
> As the reviewer correctly notes, our theory applies directly to Extreme Learning Machines (ELM) or random feature models. In these cases, the weights of the final layer correspond to the vector $w$ in our analysis.
>
> Extending the theory to standard multi-layer Neural Networks (NNs) is more challenging. In principle, for NNs with polynomial activations (which can approximate smooth activation functions like tanh), one could derive an inequality similar to our Theorem 3.5, as outlined in the proof strategy at the end of this response. However, we identify two significant challenges at this stage:
>
> * The system of equations defining the affine variety $V$ would consist of highly complex, high-degree polynomials. This makes the numerical computation of $d\_V$ computationally prohibitive, thereby hindering both theoretical validation and practical application.
>
> * Recent research indicates that the generalization of over-parameterized models like NNs is influenced not only by the geometry of the hypothesis space but also significantly by the implicit bias of the optimization algorithm. Consequently, a bound like our Theorem 3.5, which solely considers the geometry of the hypothesis space, may exhibit a significant gap from empirically observed performance.
>
> > Q2: What is the practical outcome of Theorem 3.5? Can the authors clarify, based on the aforementioned theorem, what should be good practices for building/training physics-informed models?
>
> The primary practical outcome of Theorem 3.5 is the use of the affine variety dimension, $d_V$, as a guiding principle for the efficient selection of basis/trial functions.
>
> For instance, in large spatio-temporal domains where using a dense set of collocation points (a common practice in PINNs) is computationally infeasible, $d_V$ can serve as a metric to guide an effective and tractable placement of these points prior to training.
>
> As a promising future direction, our framework suggests an alternative to methods like weak PINNs where trial functions are learned simultaneously with the main solution. Instead, one could first optimize the set of trial functions by explicitly minimizing $d_V$ before the main training process begins.
>
> > Q3: For the case of PINNs, the training is typically performed on a dense set of collocation points, in order to cover properly the whole training domain. How relevant is ensuring good generalization in these scenarios?
>
> Our theory is applicable regardless of whether the set of collocation points is dense or sparse. It provides a formal upper bound on the generalization error and helps determine the amount of training data, $n$, required to guarantee a desired performance level.
>
> As we state on lines 188-194, when the collocation points are sufficiently dense, we expect the information from the differential operator $\mathscr{D}$ to be maximally utilized. In our framework, this corresponds to a configuration that minimizes the intrinsic dimension of the solution space, i.e., $d\_{\\mathcal{V}} = \\min\_{\\mathcal{T}} \\dim(\\mathcal{V}(\\mathscr{D}, \\mathcal{B},\\mathcal{T}))$.
>
> ---
>
> ## Outline of the Proof Strategy for NNs
>
> The goal is to derive a generalization bound similar to our Theorem 3.5 for the hypothesis space $\mathcal{H}\_L$ of an NN by controlling its Rademacher complexity. While the overall framework is similar to that for linear models, we introduce a key extension to handle the non-linear, hierarchical structure of NNs.
>
> * **Step 1: Reduction from Loss Class to Hypothesis Class**
> Following standard learning theory, the generalization error is bounded by the Rademacher complexity of the loss class, $\mathfrak{R}(\ell \circ \mathcal{H}\_L)$. We then apply the Contraction Principle, leveraging the Lipschitz property of the loss function, reducing the problem to bounding the Rademacher complexity of the hypothesis class itself, $\mathfrak{R}(\mathcal{H}\_L)$. This step is common to both linear models and NNs.
>
> * **Step 2: Bounding Hypothesis Complexity via Dudley's Integral**
> Next, we bound  $\mathfrak{R}(\mathcal{H}\_L)$ using Dudley's integral theorem. This theorem bounds the complexity via an integral over the covering numbers of the hypothesis space, $\mathcal{N}(\mathcal{H}\_L, \varepsilon, \\|\cdot\\|\_{\infty})$. This step represents the key departure from the analysis of linear models.
>
> * **Step 3: Bounding the NN's Covering Number**
> Directly evaluating the covering number for a complex function space like that of an NN is often intractable. Our approach is to bound the covering number of the function space by that of the more tractable parameter space.
>    * **Stability Analysis:** Through a recursive, layer-by-layer stability analysis, we relate distances in the parameter space (weights) to distances in the function space. Specifically, we show that if two parameter vectors $w$ and $w'$ are $\varepsilon$-close, their corresponding functions $f\_w$ and $f\_{w'}$ are bounded by $\\|f\_w - f\_{w'}\\|\_{\\infty} \\le \\ell\_L \\varepsilon$. The stability constant $\\ell\_L$ depends on the network's architecture (e.g., depth $L$ and weight norm $R$).
>     * **Covering Number Relation:** This stability analysis yields the critical inequality $\\mathcal{N}(\\mathcal{H}\_L, \\ell\_L \\varepsilon, \\|\\cdot\\|\_{\\infty}) \\le \\mathcal{N}(V\_R, \\varepsilon, \\|\\cdot\\|\_2)$, which bounds the function space's covering number by the parameter space's.
>
> * **Step 4: Leveraging the Geometry of the Parameter Space**
> This reduces the problem to bounding the covering number of the parameter set $\mathcal{V}\_R$ by applying the Lemma B.2.
>
> * **Step 5: Final Assembly**
> Finally, we substitute the covering number bound from Step 4 into the Dudley integral from Step 2 (accounting for the scaling by the stability constant $\ell_L$). Solving this integral yields a concrete upper bound on $\mathfrak{R}(\mathcal{H}\_L)$ in terms of $d\_V$ and $\ell\_L$. This result is then used in the generalization inequality from Step 1 to complete the proof of the final theorem for PINN (similar results of Theorem 3.5).
>
> ## Additional Key Assumptions
> This extension requires two key assumptions:
> * Lipschitz continuity and boundedness of the activation functions.
> * A norm constraint on each layer's weight matrix.

---

> > ### Comment · Reviewer_qf6r · 2025-08-05
> >
> > I deeply appreciate the effort in the authors' rebuttal to address all my concerns. In principle, all my major concerns are properly addressed. However, I would like to ask an additional clarification to the authors: speaking of the extension to extreme learning machines (ELM), I see that the main difference with PILR would be that in the first case the approximation of the PDE solution is done through a linear combination of basis functions. However, in the case of ELM, the approximation is not supported through an orthogonal basis, but rather to a "randomized" based. How would this impact the value of $d_{\mathcal{V}}$ in practice?
> >
> > Lastly I would like the authors to clarify their intent of considering our discussion for the revised version of the manuscript. Are the authors planning to also include the sketch of the NN proof in the revised manuscript?

---

> > > ### Author Response · Authors · 2025-08-06
> > > **Response to Reviewer qf6r**
> > >
> > > > Q1. However, in the case of ELM, the approximation is not supported through an orthogonal basis, but rather to a "randomized" based. How would this impact the value of $d_{\mathcal{V}}$ in practice?
> > >
> > > ​Thank you very much for your valuable additional questions.
> > >
> > > In practice, the performance of an ELM largely depends on the quality of its chosen basis, leading to two possible scenarios.
> > > * **When the random basis provides a good approximation:** If the randomly sampled basis functions can sufficiently approximate the target function, the PDE constraint works effectively. This is the situation where, as our theory asserts, you can expect good generalization performance due to a small $d_V$. In this case, the value of $d_V$ shouldn't change significantly, even if the chosen basis set varies slightly. The critical factor isn't whether the basis is orthogonal, but whether it can approximate the target function.
> > >
> > > * **When the random basis provides a poor approximation:** On the other hand, if the sampled basis can't represent the target function well, the performance will be limited no matter how the training is done. This is because a misspecified basis introduces a large approximation error (the inability of the hypothesis space to represent the true solution), which becomes the dominant factor over the estimation error that our theory bounds.
> > > The decomposition of the total error is defined in Appendix C.4, Eq. (14):
> > > $$
> > > \\|f^* - \hat{f} \\| \leq \\underbrace{\\|f^* - f^*\_{\\mathcal{H}} \\|}\_{\\text{approximation error}} + \\underbrace{\\|f\_{\\mathcal{H}} - \\hat{f} \\|}\_{\\text{estimation error}},
> > > $$
> > >
> > > This discussion is also similar to the one about missing frequencies (Q6) with reviewer gn41. I plan to add a new subsection on the discussion of cases where this basis is not well-chosen in main body, and I will mention ELM as one example there.
> > >
> > > > Q2. Are the authors planning to also include the sketch of the NN proof in the revised manuscript?
> > >
> > > Yes, absolutely. We believe this is a very fruitful discussion, so we will add a new section to the Appendix to discuss it, including the current proof and its limitations. (Due to page constraints, it's difficult to include this in the main body.)

---

> > > > ### Comment · Reviewer_qf6r · 2025-08-06
> > > >
> > > > I kindly thank the authors for their thorough response. All my concerns have been fully addressed, and the outcomes of our (and the other reviewer's discussion) shed some interesting additional insights to the paper's results. I will happily increase my score.

---

> > > > > ### Author Response · Authors · 2025-08-06
> > > > > **Response to Reviewer qf6r**
> > > > >
> > > > > Thank you very much for your thoughtful review and for taking the time to discuss our revisions. We’re delighted to hear that all your concerns have been addressed and that the additional insights from our discussion have been helpful.

---

### Official Review · Reviewer_6adr · 2025-07-02

**Clarity:** 2
**Significance:** 2
**Originality:** 3
**Rating:** 4
**Confidence:** 2

**Summary:**

This paper analyzes the generalization of physics-informed learning through affine variety learning. Although the number of model parameters can be large, leading to poor generalization, the physical (differential equation) constraint largely restricts the hypothesis space -- for example, the 1D harmonic oscillator example only has $d_v=2$, avoiding a generalization error dependent on the number of model parameters. They propose a numerical method to estimate $d_v$, and apply the method to analyze two linear systems in order to verify the correctness of their theory.

**Questions:**

* Line 156, what is $d$ ("ambient input dimension")? For example, when we use a 1000-parameter PINN to solve a 2D PDE, is $d=2$ or $d=1000$? If $d=2$, then the situation $d_v\ll d$ becomes impossible.
* Line 307, the diffusion coefficient is 0?
* I'm slightly confused by the goal of Physics-Informed Linear Regression (PILR). Is it a theoretical framework or a training method? Section 3 seems to frame it as a theory, while Section 5 seems to suggest that it is a training method.
* Can you give a simple nonlinear example in action?

**Ethical Concerns:**

["NO or VERY MINOR ethics concerns only"]

**Final Justification:**

Thanks to the authors for addressing my questions. I would love to raise the score to 4.

**Limitations:**

Yes

**Quality:**

2

**Strengths And Weaknesses:**

Strengths:
* This paper is well-formatted and looks professional (I did not see typos, but I also disclaim that I do not understand all theorems. See questions)
* Understanding the generalization behavior of physics-informed learning (especially the benefit of physical constraints) is an important topic

Weaknesses:
* I found this paper mathematically dense. I think that some parts can be dropped without compromising the main idea. For example, I am confused by $\beta$. My best attempt to understand it is that it is related to topology due to the nonlinear differential operators, but otherwise it remains mysterious. In fact, both examples in the experiment section are linear, making the topological analysis unnecessary.

---

> ### Author Rebuttal · Authors · 2025-07-29
>
> We would like to express our sincere gratitude for your time and effort in reviewing our manuscript. Your comments have been very helpful.
>
> > Q1: Line 156, what is d ("ambient input dimension")? For example, when we use a 1000-parameter PINN to solve a 2D PDE, is d=2 or d=1000?
>
> $d$ is the number of parameters in the model, not the dimension of the PDE's domain. So in your example, $d=1000$.
>
> > Q2: Line 307, the diffusion coefficient is 0?
>
> We apologize for this typo. The correct equation is:
>
> $$
> \frac{\partial u}{\partial t} - c \frac{\partial^2 u}{\partial x^2} = 0.
> $$
>
> We will correct this in the revised manuscript.
>
> > Q3:  I'm slightly confused by the goal of PILR. Is it a theoretical framework or a training method? Section 3 seems to frame it as a theory, while Section 5 seems to suggest that it is a training method.
>
> The goal of our paper is to provide a theoretical framework for analyzing the learning problem we refer to as PILR. Section 3 details this theoretical analysis, while the purpose of Section 5 is to empirically validate our theory, not to propose a new training method. In fact, the algorithms used to solve the PILR problem in Section 5 are existing, standard methods: the analytical solution from [16] for linear cases and Adam optimizer for nonlinear cases (see lines 283-287).
>
> To prevent this misunderstanding in the manuscript, we will revise the text to clarify this distinction. The revised sentence at the beginning of the experimental section will read:
>
> Line278: To validate our theoretical analysis, our experiments compare the generalization performance of two fundamental approaches to learning the target function $f^*$: standard Ridge Regression (RR) with the fixed basis $\\mathcal{B}$, which uses only observational data, and Physics-Informed Linear Regression (PILR), which additionally incorporates constraints from physical laws.
>
> > Q4: Can you give a simple nonlinear example in action?
>
> Yes. We conducted experiments with two nonlinear equations: the nonlinear Bernoulli equation and the nonlinear diffusion equation in Section 5.
> * For the Bernoulli equation, the nonlinear term is $Qy^{\\rho}_{\tau}$, where we set $\rho = 2$.
> * For the diffusion equation, the diffusion coefficient depends on the solution: $c(u) = 0.1/(1 + u^2)$
>
> Our experiments confirm that even in these nonlinear cases, the improvement in generalization error is well-explained by the dimension of the resulting affine variety, $d_V$.
>
> > Q5. I think that some parts can be dropped without compromising the main idea. For example, I am confused by $\beta$. My best attempt to understand it is that it is related to topology due to the nonlinear differential operators, but otherwise it remains mysterious.
>
> We understand that $\beta$ is an abstract concept, but it is a crucial component of our work.
>
> A key strength of our theory is its applicability to nonlinear differential equations, which makes the topological analysis involving $\beta$ indispensable. To clarify its role, we provide several explanations in the manuscript:
>
> * A concise definition in Lines 159-165.
> * A discussion of how $\beta$ appears in the covering number upper bound in Lines 166-177.
> * A detailed, intuitive explanation with examples in Appendix B.2.
>
> Here is a brief summary of the intuitive explanation from Appendix B.2:
>
> Consider a circle defined by $V = \\{ (x, y) \in \mathbb{R}^2 \mid x^2 + y^2 - 1 = 0 \\}$. The ambient space dimension is $d = 2$, and the circle's intrinsic dimension is $d_V = 1$. In this 2D plane, we can consider affine subspaces $L$, such as lines (1-dimensional) and points (0-dimensional). The definition of a $(\beta, d_V)$-regular set depends on the codimension of these subspaces, which is $\text{codim}(L) = d - \dim(L)$.
>
> **Case 1: Intersection with Lines**
>
> For any line $L$ in the plane, its dimension is $\dim(L) = 1$. The codimension is therefore $\text{codim}(L) = 2 - 1 = 1$. Since $\text{codim}(L) \leq d_V$ (as $1 \leq 1$), this falls under the first case of the definition. Here, $\beta$ is defined as the maximum number of path-connected components of the intersection $V \cap L$.
>
> Let's see what happens when we intersect our circle $V$ with a line $L$:
> * **No intersection**: If the line misses the circle, $V \cap L$ is empty (0 components).
> * **Tangent line**: If the line touches the circle at a single point, $V \cap L$ is a single point (1 path-connected component).
> * **Secant line**: If the line cuts through the circle, $V \cap L$ consists of two distinct points. Since these points are separated, this set has two path-connected components.
>
> The maximum number of path-connected components we can get is **2**. Therefore, for the circle, $\beta = 2$.
> Intuitively, $\beta$ is related to the topology of the variety $V$. A line can intersect the single "loop" of a circle at most twice.
>
> **Case 2: Intersection with Points**
>
> Now, consider a point $L$ in the plane. Its dimension is $\dim(L) = 0$. The codimension is therefore $\text{codim}(L) = 2 - 0 = 2$.
>
> Since $\text{codim}(L) > d_V$ (as $2 > 1$), this falls under the second condition, which requires  $V \cap L$ to be empty for "most" such subspaces. This condition essentially requires that the intersection $V \cap L$ is empty for "most" such subspaces.
> This holds for the circle, as most points in the 2D plane do not lie on the circle itself.
>
> The parameter $\beta$ is crucial because it bounds how many disconnected pieces of the variety we must cover when we intersect it with a subspace L. As our covering construction in the paper (following [41]) places the centers of ϵ-balls on these intersections $V \cap L$, $\beta$ directly influences the final covering number bounds by controlling the complexity of these intersections.
>
> [41] Yifan Zhang and Joe Kileel. Covering number of real algebraic varieties and beyond: Improved453bounds and applications. arXiv e-prints, pages arXiv–2311, 2023.454.

---

> > ### Comment · Reviewer_6adr · 2025-08-08
> >
> > I would like to thank the authors for their clarification, which addresses most of my concerns. Therefore, I'm raising my score.

---

### Official Review · Reviewer_gn41 · 2025-07-03

**Clarity:** 2
**Significance:** 3
**Originality:** 3
**Rating:** 4
**Confidence:** 3

**Summary:**

The paper proposes a theoretical framework for physics-informed linear regression. They theoratically show that generalization cap is governed by the dimension of an affine variety induced by physical constraints.

They also naturally derive from this assertion to the linear operator case (the rank-nullity theorem), which is commonly used in traditional linear ROM and galerkin methods.

The unified weak-form formulation and minimax risk analysis on the affine variety are novel to the nonlinear operator cases, and experiments on 3 benchmark PDEs demonstrate the effectiveness of their theory.

My overall take: paper is well writen with minor improvements for clarity. I recommend boarderline accept and look forward to revision.

**Questions:**

1. How are \((\psi_k, \mu_k)\) chosen for later 2 cases? For the 1st case only Dirac measures are used, does the theory still hold for general \(\mu_k\)? (maybe an ablation study?)
2. Can the framework handle *unknown terms in \(\mathcal{D}\)* (not just \(v/g\))? If no, any thoughts?
3. I am also curious how does \(d_{\mathcal{V}}\) (and the final performance) change if the basis \(\{\phi_j\}\) is misspecified (e.g., missing key frequencies)?  You mentioned mis-used basis can downgrade performance, maybe show an ablation study?

**Ethical Concerns:**

["NO or VERY MINOR ethics concerns only"]

**Final Justification:**

No score adjustment; authors agreed to incorperate my suggestions; the reply is satisfactory; the current shape of manuscript is worth boarderline accept. Hence I kept my score.

**Limitations:**

No; the authors mention many good applications/extensions, however miss this one.

Although they "claimed" this method can handle partial known physics, by their Appendix C, this actually refers to unknown source term or boundary value. What if some term(s) in $\mathcal{D}$ is also unknown? How would they like to approach this?

**Paper Formatting Concerns:**

NA;

**Quality:**

3

**Strengths And Weaknesses:**

### **Strengths**
- The minimax bound and connection to kernel methods are solid advances for linear physics-informed models.
- The discrete weak form unifies collocation and variational constraints.
- Experiments confirm that smaller \(d_{\mathcal{V}}\) correlates with better generalization (Section 5).

### **Weaknesses & Clarifications Needed**
#### **1. Inconsistent Experimental Setup**
- The later 2 experiments do not specify the trial functions \((\psi_k)\) or measures \((\mu_k)\), despite their theoretical importance.
- Also, for the 1st experiment, the authors choose collocation (\(\mu_k = \delta_{x_k}\)), which is actually the strong form and contradicts the general weak-form claims.
- **Request**: Clarify the choice of \((\psi_k, \mu_k)\) for all experiments. If Dirac measures are used, justify why this is a good choice (eg, by ablation study).

#### **2. Misleading "Partial Knowledge" Description**
- The paper implies handling incomplete physics (e.g., unknown terms in PDEs) in its abstract and introduction, but Appendix C reveals the "unknown partial knowledge" is merely *source terms \(v\)* or *boundary conditions \(g\)* (not \(\mathcal{D}\) itself). This reads misleading.
- **Request**: Revise the abstract/introduction to clearly clarify that \(\mathcal{D}\) is assumed fully known. If the method cannot handle missing terms in \(\mathcal{D}\), state this limitation explicitly in future works.

#### **3. Fixed Basis Misrepresentation**
- Through reading and many terms like "data-driven", it's not clear if the basis \(\{\phi_j\}\) is fixed or learned. This only becomes confirmed after reading the studied PDEs and the chosen basis for them. Maybe mention it earlier in methodology section.

---

> ### Author Rebuttal · Authors · 2025-07-29
>
> Thank you very much for your careful review of our submission. We highly value your constructive feedback and suggestions.
>
> > Q1. How are $(\psi\_k, \mu\_k)$ chosen for later 2 cases?
>
> The explicit definitions for the trial functions and measures used in these experiments are provided in Appendix G.2, as mentioned on line 330. These are written right after line 804 and right after line 818.
>
> > Q2. For the 1st case only Dirac measures are used, does the theory still hold for general (\mu_k)? If Dirac measures are used, justify why this is a good choice?
>
> We thank the reviewer for this insightful question. The reviewer is correct that our original experiments focused heavily on the Dirac measure. To demonstrate that our theory holds for general measures, we have conducted an additional experiment for the Harmonic Oscillator using sine and cosine test functions (i.e., $\mathcal{T}$ includes basis functions from $\mathcal{B}$) with the Borel measure.
>
> In this case, we obtain $d\_V=2$, just as with the Dirac measure. Consequently, the results are nearly identical to those originally shown in Figure 1 (a). The new results are as follows:
> | d | 5 | 9 | 17 | 33 |
> | --- | --- | --- | --- | --- |
> | PILR(n=10) | 0.012 ± 0.002 | 0.012 ± 0.002 | 0.012 ± 0.002 | 0.012 ± 0.002 |
> | RR (n=10) | 0.014 ± 0.002 | 0.031 ± 0.019 | 1.015 ± 0.433 | 1.187 ± 0.525 |
> | PILR(n=20) | 0.011 ± 0.002 | 0.011 ± 0.002 | 0.011 ± 0.002 | 0.011 ± 0.002 |
> | RR (n=20) | 0.012 ± 0.002 | 0.014 ± 0.003 | 0.071 ± 0.055 | 1.002 ± 0.479 |
> | PILR(n=40) | 0.010 ± 0.001 | 0.011 ± 0.001 | 0.011 ± 0.001 | 0.011 ± 0.001 |
> | RR (n=40) | 0.011 ± 0.002 | 0.012 ± 0.002 | 0.016 ± 0.005 | 0.118 ± 0.123 |
>
> These results confirm that our core findings are robust to the choice of measure. We will add this result as a new figure in the Appendix.
>
> It is important to note that the goal of our paper is not to prescribe the optimal choice of trial functions or measures, but rather to demonstrate that our theoretical framework holds across a variety of settings for $\mathscr{D}$, $\mathcal{B}$, and $\mathcal{T}$.
>
> > Q3. Revise the abstract/introduction to clearly clarify that D is assumed fully known. If the method cannot handle missing terms in D, state this limitation explicitly in future works.
>
> We thank the reviewer for bringing this point to our attention. Our use of "incomplete knowledge" refers specifically to the source terms or boundary conditions, not the differential operator $\mathscr{D}$ itself, which we assume to be known, as detailed in Appendix C.
>
> To resolve this confusion, we will revise the manuscript as follows:
>
> * Line 26: prior knowledge of the governing differential equations, **particularly their source terms or boundary conditions**, is often incomplete.
> * Line 34, 90: formulate the ~~incomplete~~ differential equation constraints
> * Line 96: the discrete weak form of the **known** differential equation $\mathscr{D}$ is defined by
>
> > Q4: Can the framework handle unknown terms in D? If no, any thoughts?
>
> Yes, our framework will be readily extended to a setting where the operator $\mathscr{D}$ has unknown parameters (e.g., an unknown diffusion coefficient $c$) that are learned simultaneously from the data. For instance, in the diffusion equation example, if $c$ were also to be learned, one could simply analyze the affine variety $\mathcal{V} = \{(w, c) : \langle \mathscr{D}_c[w^\top \phi], \psi_k \rangle = 0\}$ defined by polynomials over an augmented parameter space including both the solution weights w and the operator parameters c. We will add this point to the Appendix as a direction for future work.
>
> However, to improve the main paper's clarity and focus, we will remove the ambiguous "incomplete" description regarding $\mathscr{D}$ and explicitly state that the operator is assumed to be known, as detailed in our response to Q3.
>
> > Q5: Through reading and many terms like "data-driven", it's not clear if the basis is fixed or learned.
>
> The basis functions are fixed, not learned, as part of the modeling process. To clarify this, we will revise line 104 as follows:
>
> Line 104: "Let B be a fixed basis (i.e., not learned like a neural network)."
>
> > Q6: I am also curious how does $d_V$ (and the final performance) change if the basis $\phi_j$ is misspecified (e.g., missing key frequencies)? You mentioned mis-used basis can downgrade performance, maybe show an ablation study?
>
> This is an excellent question. We have performed the requested ablation study for the Harmonic Oscillator. In this experiment, the basis $\\{\phi\_j\\}$ is intentionally misspecified by omitting the known frequency of the analytical solution. We used the Dirac measure and a constant trial function, with $K := |\mathcal{T}|$ set sufficiently large. The results are below:
>
> | total error / approx_error | d=17 | d=33 |
> | --- | --- | --- |
> | PILR (n=10) | 1.435 ± 0.646/1.430 ± 0.645 | 1.434 ± 0.646/1.430 ± 0.645 |
> | PILR (n=40) | 1.435 ± 0.646/1.430 ± 0.645 | 1.435 ± 0.646/1.430 ± 0.645|
>
> The results show that performance is poor and does not improve with more data $n$ and less dimension $d$. This is because a misspecified basis introduces a large approximation error (the inability of the hypothesis space to represent the true solution), which becomes the dominant factor over the estimation error that our theory bounds.
>
> The decomposition of the total error is defined in Appendix C.4, Eq. (14):
>
> $$
> \\|f^* - \hat{f} \\| \leq \\underbrace{\\|f^* - f^*\_{\\mathcal{H}} \\|}\_{\\text{approximation error}} + \\underbrace{\\|f\_{\\mathcal{H}} - \\hat{f} \\|}\_{\\text{estimation error}},
> $$
>
> where $ f^* $    is the true function and $f^*\_{\\mathcal{H}}$ is the best possible approximation within the hypothesis $\\mathcal{H} = \\{ w^{\\top} \\phi : w \\in \\mathcal{V}\_R \\}$.
>
> In this misspecified case, the approximation error term is large and irreducible. The physically constrained solution space may not contain any non-trivial solution, leading to $d_V=0$. We will add this experiment and discussion to the Appendix.

---

> > ### Comment · Reviewer_gn41 · 2025-08-02
> > **reply 1**
> >
> > Dear authors, thanks a lot for your responses.
> >
> > Here are my followups.
> >
> > I would response 1. and 2. together, as they are connected.
> > - As in my original comment, the clarity is the main issue of this paper. In abstract, authors claim "To address these limitations, we introduce a discrete weak form that unifies collocation and variational methods,..."; however, in real experiments, we can see using collocation is no longer weak form at all. This makes the abstract misleading. In response 2, authors mentioned "The reviewer is correct that our original experiments focused heavily on the Dirac measure. To demonstrate...", stating indeed the generalization to other basis are added later and using a different setting (also indicating the misleading abstract is modified later).
> > - As a result, the clarity issue for the experiments arrangement: some parts are using collocation, some are using variational, no clear illustration, just refer readers to jump to appendix. This is not a responsible writing style. **Adding more appendix does not solve this issue. And if the overall writing becomes chaotic for future readers, I may re-consider my score.**
> > - Can you clearly trim your abstract to remove ambiguitious claims? and re-arrange experiments, eg, sth like seperate sections for testing different measures for all datasets; if the experiments using variational basis are not for all cases, you should clearly list them as an extension, with clear sub-section names and writings in the main text.
> >
> > 3. Thank you so much. That should be the spirit for clear writing. Hopefully you can do that to 1. and 2. A side note, a good writing should make contents as clear as possible in main text, with that, you do not need to dump repeated information again to appendix; (This also applies to my all other writing suggestions).
> >
> > 4. Thanks for discussing that. Coefficients is okay, what about terms, eg, I do not know if it's a linear diffusion term? You are welcome to add follow up discussions in your future work (do it in the end of main text, conclusion and fusion work, not appendix)
> >
> > 5. Thanks for revision
> >
> > 6. Thanks for follow-up experiments;
> > - first this raise a key question/limitation to add: how to know and choose the key frequencies pre-hand applying your method? in a realistic application, we do not have analytical solutions prehand?
> > - if you have a solution to estimate the proper basis to use, can you discuss it here, and better apply it to show your method is practical (ie, not limited to toy cases where you already know analytical solutions and can choose perfect basis pre-hand)?
> > - secondly, this (experiments + discussion) should be put in the main text, the table should also add the results with correct basis to clearly show "...(without key frequencies), perfromance is becoming poor". If combined with the additional discussion I suggested, this can be a good sub section in sum concerning realistic settings (do not know analytical solutions pre-hand).

---

> > > ### Author Response · Authors · 2025-08-03
> > > **Response to Reviewer gn41 (1/2)**
> > >
> > > I sincerely apologize for the inadequacy of my response in the rebuttal. I also deeply appreciate your thoughtful and renewed feedback.
> > >
> > > > Q1. Can you clearly trim your abstract to remove ambiguitious claims?
> > >
> > > We apologize for the lack of clarity on this point and thank you for pointing it out again. Based on your feedback, we will make the following two main revisions:
> > >
> > > * **Change in Terminology**: We will replace the potentially misleading term "discrete weak form" with a new term, "**Unified Residual Form**."
> > > * **Clarification of Explanation**: We will explicitly state that the Unified Residual Form is not a strict strong or classical weak solution (which requires an infinite number of trial functions) but rather a practical framework for **approximating** them. We will re-emphasize that it encompasses both the use of Dirac measures (for collocation methods) and Borel measures (for variational methods).
> > >
> > > Below are the proposed revisions, particularly those that enhance the explanation:
> > > * Line 6: To address these limitations, we introduce **a Unified Residual Form** that provides a common framework for collocation and variational methods, enabling ...
> > > * Line 33: Our key idea is to formulate the differential equation constraints by introducing **a Unified Residual Form**. **This form, defined on a finite set of trial functions and a measure, provides a practical approximation of physical constraints.**
> > > * Line 99: This formulation relaxes the classical smoothness requirements while still leveraging physics-informed constraints via a unified measure-based approach: choosing Borel measures leads to **an approximation of standard weak solutions**, whereas choosing Dirac measures leads to **an approximation of the strong-form residuals** used in the PINN framework.
> > >
> > > > Q2. Can you re-arrange experiments, eg, sth like seperate sections for testing different measures for all datasets; if the experiments using variational basis are not for all cases, you should clearly list them as an extension, with clear sub-section names and writings in the main text.
> > >
> > > Thank you for your very constructive suggestion.
> > >
> > > To improve the clarity of the experimental setup, we will make the following changes:
> > > * We will add a new subsection, **Section 5.2 Learning Weak Solutions**, and include the results for the Harmonic Oscillator and the Diffusion Equation.  (We are currently conducting additional experiments on the diffusion equation and will share the results during the discussion period if they are ready.)  We will add graphs for these results, similar to Figure 1(a) and (b). Furthermore, for the sake of clarity, the following sentence will be inserted at the beginning of Sections 5.1 and 5.2.
> > >     * (At the beginning of Section 5.1): In this section, we investigate the strong solutions, ..., **by employing the Dirac measure, which corresponds to the collocation method used in PINNs**.
> > >      * (At the beginning of Section 5.2): In this section, we learn the weak solutions, ..., **by employing the Borel measure, consistent with variational formulations.**
> > >
> > >
> > > * In **Section 5.3 Learning Numerical Solutions**, we will explicitly state at the beginning of the section that the experiments use the Dirac measure. Furthermore, we will specify the basis functions, trial functions, and measures used in each experiment directly in the main text.
> > >
> > >
> > > Here are the more specific revisions for Section 5.3:
> > > * Line 318: In this section, we use piecewise constant functions corresponding to the FDM grid as basis functions, the Dirac measure on the grid points, and a constant function over the entire domain as the trial function.
> > > * Line 324: The settings for the basis functions and the trial functions with measure $\phi_{\tau} \in \mathcal{B}_h$, $(\psi\_{\tau}, \mu\_{\tau}) \in \mathcal{T}\_h$ are as follows: ... (from Line 804)
> > > * Line 329: The settings for the basis functions and the trial functions with measure $\phi_{j, \tau} \in \mathcal{B}\_{h},\ (\psi\_{j, \tau}, \mu\_{j, \tau}) \in \mathcal{T}\_{h}$ are as follows: ... (from Line 818)
> > >
> > > > Q3. Coefficients is okay, what about terms, eg, I do not know if it's a linear diffusion term?
> > >
> > > Thank you for this sharp observation. In this case, we can introduce a new linear model, $v_{\theta}$, parameterized by $\theta$, and for the equation $\partial_t u_{w} + c v_{\theta} = 0$, we can consider the affine variety in the $(w, \theta)$ space.
> > >
> > > We will add the following statement to the conclusion as a direction for future work:
> > > * As a future extension, for differential equations with unknown parameters or terms, one could consider a method where the unknown parts are parameterized by learnable parameters $\theta$, and the solution weights $w$ and parameters $\theta$ are learned simultaneously. In this case, our theory can still be applied by considering the affine variety in the extended parameter space $(w, \theta)$.
> > >
> > > ---
> > > Due to space limitations, the continuation is provided in the following thread.

---

> > > > ### Author Response · Authors · 2025-08-03
> > > > **Response to Reviewer gn41 (2/2)**
> > > >
> > > > > Q4. how to know and choose the key frequencies pre-hand applying your method? in a realistic application, we do not have analytical solutions prehand?
> > > >
> > > > Thank you for your insightful comment.
> > > >
> > > > However, providing a definitive guideline on “how to design a basis that yields an accurate approximation while keeping the generalization complexity $d_V$ small” is essentially a **model-selection problem** and therefore remains an open avenue for future work.
> > > >
> > > > The theory developed in this study is intended to explain why imposing differential-equation constraints on a given basis can (or cannot) achieve high accuracy. In other words, it offers a theoretical framework for understanding why “using a good basis together with differential-equation constraints improves accuracy” (and conversely, why it might fail).
> > > >
> > > > While indices such as the $d_V$ derived here could in principle enable more efficient selection of bases (model architectures), it is still unknown how effective they would be in practice compared with standard techniques such as cross-validation; this, too, is a topic for future investigation.
> > > >
> > > > Because $d_V$ is a discrete dimensional index, it is conceptually useful for theoretical understanding but less convenient in settings that require continuous optimization or practical architecture selection. Future research may therefore need to propose smoother, more optimization-friendly indices related to $d_V$.
> > > >
> > > > The difficulty of deriving direct design guidelines from generalization theory for architectural choices is not specific to the present work; it is a limitation shared across generalization theory as a whole. Bridging theory and concrete design will require additional theoretical and empirical studies.
> > > >
> > > > We will add the following limitation to Conclusion Section:
> > > > * The establishment of a systematic methodology that translates the dimensional $d_V$ or related theoretical indicators into actionable guidelines for the practical design of basis and trial functions remains a major unresolved challenge.
> > > >
> > > > > Q5. secondly, this (experiments + discussion) should be put in the main text, the table should also add the results with correct basis to clearly show "...(without key frequencies
> > > >
> > > > Thank you for the suggestion. We have newly added Section 5.4 to the main paper, in which we discuss cases where the basis was not appropriately chosen and present the results of additional experiments introduced in our rebuttal, including the corresponding result tables.
> > > >
> > > > ---
> > > >
> > > > We sincerely apologize for having concentrated many of the revisions in the appendix, as we had overlooked the fact that one additional page was allowed in the accepted version of the paper.

---

> ### Author Response · Authors · 2025-08-05
> **Additional Results for the diffusion equation & the manuscripts for Section 5.2**
>
> I've completed the experimental results and the detail draft, and I'd like to share them with you.
>
> The experimental results for the diffusion equation, which correspond to fig:diff-perf-weak in main texts, are shown below. For details on the experimental setup and other information, please refer to the manuscripts of Section 5.2.
> The experimental results shown in fig:ho-perf-weak correspond to the weak solution of the harmonic oscillator. I have already provided these results in my rebuttal response to Q2.
>
> **Results for weak solution of diffusion equation**
>
> | method \ d_x | 10 | 15 | 20 | 25 |
> | --- | --- | --- | --- | --- |
> | PILR(n=20) | 0.039 ± 0.009 | 0.042 ± 0.010 | 0.039 ± 0.012 | 0.039 ± 0.012 |
> | RR (n=20) | 0.141 ± 0.036 | 0.275 ± 0.148 | 0.368 ± 0.148 | 0.499 ± 0.186 |
> | PILR(n=40) | 0.020 ± 0.006 | 0.021 ± 0.006 | 0.022 ± 0.005 | 0.022 ± 0.006 |
> | RR (n=40) | 0.033 ± 0.013 | 0.057 ± 0.023 | 0.204 ± 0.085 | 0.254 ± 0.125 |
> | PILR(n=60) | 0.017 ± 0.005 | 0.016 ± 0.003 | 0.017 ± 0.003 | 0.017 ± 0.004 |
> | RR (n=60) | 0.019 ± 0.005 | 0.031 ± 0.012 | 0.068 ± 0.032 | 0.150 ± 0.065 |
>
> **Manuscripts of Section 5.2 Learning Weak Solutions**
>
> ---
>
> \subsection{Learning Weak Solutions}
>
> In this section, we investigate weak solutions for the harmonic oscillator and the diffusion equation, by employing the Dirac measure, which corresponds to the collocation method used in PINNs. The governing equations and basis functions are identical to those in Section 5.1.
>
> \paragraph{Harmonic Oscillator}
>
> We define the trial functions $\psi\_k$ for $1 \leq k \leq K\_t$ as:
>
> $$
> \\psi_1(x) = 1, \\quad \\psi_{2k-1}(x) = \\cos(\\omega_k x), \\quad \\psi_{2k}(x) = \\sin(\\omega_k x),
> $$
>
> where $\omega_k \coloneqq \frac{k\pi}{T}$ is the frequency. The associated measure $\mu_k$ is the Lebesgue measure on $\Omega = [0, T]$.
>
> The dimension of the affine variety remains $d_{\mathcal{V}} = 2$, consistent with the strong solutions. Experimental results in \cref{fig:ho-perf-weak} confirm this. The performance of PILR is stable and independent of the number of basis functions, unlike RR, which shows performance degradation as model complexity increases.
>
> \paragraph{Diffusion Equation}
>
> The trial functions $\psi_{k, k'}$ combine a piecewise constant basis in time and a Fourier basis in space. For indices $1 \leq k \leq K_t$ and $1 \leq k' \leq K_x$, they are:
>
> $$
> \\psi\_{k, 2k'-1}(x,t) = 1\_{[t\_k, t\_{k+1}]}(t) \\cos(\\omega\_{k'} x), \\quad \\psi\_{k, 2k'}(x,t) = 1\_{[t\_k, t\_{k+1}]}(t) \\sin(\\omega\_{k'} x),
> $$
> where $1_{[t_k, t_{k+1}]}(t)$ is the indicator function for the time interval, defined as
> $$
> 1\_{[t\_k, t\_{k+1}]}(t) =
> \\begin{cases}
>     1 & \\text{if } t \\in [t\_k, t\_{k+1}) \\\\
>     0 & \\text{otherwise}
> \\end{cases},
> $$
>
> and $\omega_{k'} = \frac{k'\pi}{\Xi}$. The associated measure $\mu_k$ is the Lebesgue measure on $[-\Xi, \Xi] \times [0, T]$.
>
> The dimension of the affine variety, $d_{\mathcal{V}} = 2 \min(d_x, d_t) + 1$, is identical to the strong solution case. The results in \cref{fig:diff-perf-weak} show that PILR's generalization performance remains robust as the number of spatial basis functions $d_x$ increases, demonstrating its advantage over RR.
>
> ---

---

> > ### Comment · Reviewer_gn41 · 2025-08-06
> > **reply 2**
> >
> > Dear authors, I have checked all the updated results and revisions you provided.
> >
> > Overall, these are good! I sincerely hope that you incorperated these revisions in your final manuscript.
> >
> > I will maintain my original score as it's already fair (and boarderline acceptance).

---

> > > ### Author Response · Authors · 2025-08-07
> > > **Response to gn41**
> > >
> > > Thank you very much for your time and effort in reviewing our manuscript. We truly appreciate your insightful discussion and feedback, and we will be sure to incorporate all of the revisions.

---

### Official Review · Reviewer_sUwm · 2025-07-06

**Clarity:** 4
**Significance:** 4
**Originality:** 3
**Rating:** 5
**Confidence:** 3

**Summary:**

This paper proposes a theoretical analysis to study the generalisation of physics-informed linear regression problems where the physical information is introduced in the form of weak constraints on the hypothesis space. The authors prove a probabilistic minimax risk bound on the estimation error in terms of the intrinsic dimension dV of the affine variety associated with the physical constraints. A practical estimator for dV is proposed and experiments on linear and nonlinear systems are usedd to demonstrate the bound.

**Questions:**

- Could the author provide an example where something else than a Borel or Dirac measure for μk would be a useful choice? This is just out of personal curiosity
- Certainly due to my lack of familiarity with algebraic geometry I'm still struggling to understand what are path-connected components in a (β, dv)-regular set. Could the author provide some intuition on this?

**Ethical Concerns:**

["NO or VERY MINOR ethics concerns only"]

**Final Justification:**

I appreciated the paper, which I believe will make a significant contribution to the field, and will be happy to see it published.

**Limitations:**

yes

**Quality:**

4

**Strengths And Weaknesses:**

Strengths
- I found the theoretical analysis proposed by this paper very compelling. The analysis of the role of constraints as the dimensionality of their induced variety is elegant and provides theoretical tools to understand the generalization of PINN that matches practical experience from working with them. This is a very welcome piece of theoretical understanding which will hopefully help inform future design choices for physics-informed learning models.
- It is great to see alignment of the proposed theory with previous work on PI-kernels under linear operators, this is precisely a question I had in mind while reading section 3 and it was refreshing to see the authors address it immediately
- The author provide a practical rule of thumb to get a lower bound on dV so the theory can guide model design
- Generally, the paper has a clear exposition which doesn't overwhelm the reader with unnecessary technicalities in the main text, but refers to comprehensive theoretical background in the supplementary material (I haven't checked the proofs)
- The experiments are clear and convincing, addressing the kind of ablation questions one would typically expect. This feels like polished work.



Weaknesses
- The assumptions made in Section 3 are reasonable for theoretical analysis, but real physical systems may in practice violate them - that could be worth a mention for readers coming with a physics-background
- One thing that somewhat got me confused is that the paper claims that in contrast with prior work, they don't assume complete knowledge of the differential equation. Yet, it seems that the variety in (2) is defined in terms of the exact differential equation operator D. I suppose that "incomplete knowledge" referred to the weak form of constraints, but I thought for a second we would be assuming some sort of approximate differential operator D.

---

> ### Author Rebuttal · Authors · 2025-07-29
>
> Thank you very much for taking the time to review our manuscript. Your feedback is greatly appreciated.
>
> > Q1: The paper's assumptions are reasonable for theory but may not hold in practice.
>
> We agree that acknowledging the potential gap between our theoretical assumptions and real-world application is crucial. We will add the following statement to Section 3 to clarify this for readers:
>
> "While we assume Gaussian noise for analytical tractability, noise in observational data from physical phenomena can be far more complex. Bridging this gap is an important direction for future work."
>
> > Q2: The claim of 'incomplete knowledge' is confusing, as the operator $\mathscr{D}$ seems to be fully known.
>
> We thank the reviewer for highlighting this ambiguity. Our use of "incomplete knowledge" refers specifically to the source terms or boundary conditions, not the differential operator $\mathscr{D}$ itself, which we assume to be known, as detailed in Appendix C.
>
> To resolve this confusion, we will revise the manuscript as follows:
>
> * Line 26: prior knowledge of the governing differential equations, **particularly their source terms or boundary conditions**, is often incomplete.
> * Line 34, 90: formulate the ~~incomplete~~ differential equation constraints
> * Line 96: the discrete weak form of the **known** differential equation $\mathscr{D}$ is defined by
>
> Furthermore, we will add a note in the Appendix that our framework can be extended to cases where $\mathscr{D}$ is partially unknown (e.g., an unknown diffusion coefficient $c$). This would involve analyzing an affine variety $\mathcal{V}$ defined by polynomial constraints on an augmented parameter space that includes both the model weights $w$ and the unknown physical parameter $c$. For the linear diffusion equation, this would correspond to the variety $\mathcal{V} = \\{(w, c) : \langle \mathscr{D}_c[w^\top \phi], \psi_k \rangle = 0\\}$. We will add a note on this possible extension in the Appendix.
>
> > Q3: Could the author provide an example where something else than a Borel or Dirac measure for $\mu_k$ would be a useful choice?
>
> Yes. A key example arises when extending the problem to differential equations on a curved manifold, such as the unit sphere $\mathbb{S}^2$. In such settings, it is natural to define the integral using the Riemannian volume measure $\mathrm{d}\mathrm{Vol}_{\mathcal{M}}$. For instance, Zhou and Shi (2025) recently adopted this approach on manifolds.
>
> Zhou, Hanfei, and Lei Shi. "Weak Physics Informed Neural Networks for Geometry Compatible Hyperbolic Conservation Laws on Manifolds." arXiv preprint arXiv:2505.19036 (2025).
>
> > Q4: I'm still struggling to understand what are path-connected components in a $( \beta, d\_V )$-regular set. Could the author provide some intuition on this?
>
> We appreciate the opportunity to provide more intuition. We have a detailed explanation in Appendix B.2, using the minimal example of a circle in a 2D plane, which we summarize here.
>
> Consider a circle defined by $V = \\{ (x, y) \in \mathbb{R}^2 \mid x^2 + y^2 - 1 = 0 \\}$. The ambient space dimension is $d = 2$, and the circle's intrinsic dimension is $d_V = 1$. In this 2D plane, we can consider affine subspaces $L$, such as lines (1-dimensional) and points (0-dimensional). The definition of a $(\beta, d_V)$-regular set depends on the codimension of these subspaces, which is $\text{codim}(L) = d - \dim(L)$.
>
> **Case 1: Intersection with Lines**
>
> For any line $L$ in the plane, its dimension is $\dim(L) = 1$. The codimension is therefore $\text{codim}(L) = 2 - 1 = 1$. Since $\text{codim}(L) \leq d_V$ (as $1 \leq 1$), this falls under the first case of the definition. Here, $\beta$ is defined as the maximum number of path-connected components of the intersection $V \cap L$.
>
> Let's see what happens when we intersect our circle $V$ with a line $L$:
> * **No intersection**: If the line misses the circle, $V \cap L$ is empty (0 components).
> * **Tangent line**: If the line touches the circle at a single point, $V \cap L$ is a single point (1 path-connected component).
> * **Secant line**: If the line cuts through the circle, $V \cap L$ consists of two distinct points. Since these points are separated, this set has two path-connected components.
>
> The maximum number of path-connected components we can get is **2**. Therefore, for the circle, $\beta = 2$.
> Intuitively, $\beta$ is related to the topology of the variety $V$. A line can intersect the single "loop" of a circle at most twice.
>
> **Case 2: Intersection with Points**
>
> Now, consider a point $L$ in the plane. Its dimension is $\dim(L) = 0$. The codimension is therefore $\text{codim}(L) = 2 - 0 = 2$.
>
> Since $\text{codim}(L) > d_V$ (as $2 > 1$), this falls under the second condition, which requires  $V \cap L$ to be empty for "most" such subspaces. This condition essentially requires that the intersection $V \cap L$ is empty for "most" such subspaces.
> This holds for the circle, as most points in the 2D plane do not lie on the circle itself.
>
> The parameter $\beta$ is crucial because it bounds how many disconnected pieces of the variety we must cover when we intersect it with a subspace L. As our covering construction in the paper (following [41]) places the centers of ϵ-balls on these intersections $V \cap L$, $\beta$ directly influences the final covering number bounds by controlling the complexity of these intersections.
>
> [41] Yifan Zhang and Joe Kileel. Covering number of real algebraic varieties and beyond: Improved453bounds and applications. arXiv e-prints, pages arXiv–2311, 2023.454.

---

> ### Comment · Reviewer_sUwm · 2025-08-02
> **Response to authors**
>
> I want to thank the authors for engaging constructively in the review process, and answering my questions with an educative example. I appreciated the paper, which I believe will make a significant contribution to the field, and will be happy to see it published. I have no further comments.

---

> > ### Author Response · Authors · 2025-08-03
> > **Response to Reviewer sUwm**
> >
> > Thank you very much for your thoughtful and encouraging comments.
> > We truly appreciate your constructive engagement throughout the review process, and we are glad that our explanation and example were helpful in addressing your questions.

---

### Note · Authors · 2025-08-12

**Question Regarding Reviewer Score**

Before the final decision, could you please confirm the final rating from Reviewer qf6r? They stated, "I will happily increase my score," but their final score appears to remain unchanged, presumably by mistake.

**Summary of Discussions**

Here is a summary of the key points of discussion during the review process.

1. On Terminology: To address concerns that our key terms "incomplete knowledge" and "discrete weak form" were ambiguous, we clarified the scope of the former. We also proposed replacing the latter with "unified residual form" to better capture our framework's ability to unify both strong-form (collocation) and weak-form (variational) methods.

2. On the Experimental Setup: To resolve critiques about the experimental section's lack of clarity, we committed to a significant restructuring. This included conducting new experiments with weak-form learning to demonstrate robustness, reorganizing the section into distinct parts for clarity, and, critically, adding clear definitions of the basis and trial functions to the main body of the paper.

3. On Basis Functions: When questioned about misspecified basis functions, including Extreme Learning Machines (ELMs), we clarified that the resulting performance degradation is due to approximation error, distinct from the estimation error our analysis focuses on. We committed to adding a new subsection with experiments to demonstrate this point and explained that our theory serves as a post-hoc diagnostic tool, not an a priori guide for model selection.

4. On Practicality and Applicability: In response to questions about applicability to NNs, we explained that while extending our theoretical bounds to neural networks is conceptually feasible, it presents significant challenges. We added a discussion of this potential extension and its limitations to the appendix. Furthermore, we positioned our work as a foundational, geometric interpretation of generalization that shows a promising direction for future research on theory-guided model selection, such as choosing optimal basis and trial functions.

---
We are sincerely grateful to all the reviewers and the Area Chair for their time and constructive feedback, which have been invaluable in improving our work.

---

### Decision · Program_Chairs · 2025-09-17

**Decision:**

Accept (poster)

**Comment:**

After discussion, all reviewers centered on a positive assessment. Two main weaknesses were identified:

1. The paper made a misleading claim about not requiring knowledge of the PDE. The AC’s opinion is that this is, in general, a common criticism of PINN-type methods and not unique to this paper. However, the authors should directly address this concern in the revised manuscript and make sure to clarify what aspects of the problem need to be known for the proposed method.
2. The paper is limited to only some types of physics informed models. This is also not disqualifying.

All authors ultimately concurred that the paper would be a valuable contribution to the field. Thus, the AC finds no reason to overturn consensus and recommends accept.